# On the effects of hard and soft equality constraints in the iterative outlier elimination procedure

Vinicius Francisco Rofatto[1,2]*, Marcelo Tomio Matsuoka[1,2,3,6], Ivandro Klein[4,5], Maurício Roberto Veronez[6], Luiz Gonzaga da Silveira Junior[6]

**1** Graduate Program in Remote Sensing, Federal University of Rio Grande do Sul, Porto Alegre, RS, Brazil, **2** Institute of Geography, Federal University of Uberlandia, Monte Carmelo, MG, Brazil, **3** Graduate Program in Agriculture and Geospatial Information, Federal University of Uberlandia, Monte Carmelo, MG, Brazil, **4** Department of Civil Construction, Federal Institute of Santa Catarina, Florianópolis, SC, Brazil, **5** Graduate Program in Geodetic Sciences, Federal University of Paraná, Curitiba, PR, Brazil, **6** Graduate Program in Applied Computing, Unisinos University, São Leopoldo, RS, Brazil

* vfrofatto@gmail.com

## Abstract

Reliability analysis allows for the estimation of a system's probability of detecting and identifying outliers. Failure to identify an outlier can jeopardize the reliability level of a system. Due to its importance, outliers must be appropriately treated to ensure the normal operation of a system. System models are usually developed from certain constraints. Constraints play a central role in model precision and validity. In this work, we present a detailed investigation of the effects of the hard and soft constraints on the reliability of a measurement system model. Hard constraints represent a case in which there exist known functional relations between the unknown model parameters, whereas the soft constraints are employed where such functional relations can be slightly violated depending on their uncertainty. The results highlighted that the success rate of identifying an outlier for the case of hard constraints is larger than soft constraints. This suggested that hard constraints be used in the stage of pre-processing data for the purpose of identifying and removing possible outlying measurements. After identifying and removing possible outliers, one should set up the soft constraints to propagate their uncertainties to the model parameters during the data processing.

## Introduction

It is very common to build models (i.e., the equation systems) based on some initial knowledge about a given problem. In other words, models are often set up in a way that the model parameters need to fulfill certain constraints. Such constraints are a priori knowledge embedded into a model to avoid a trivial solution; to guarantee the stability of estimates; to improve the precision and accuracy of the results by reducing the number of unknown parameters, or

**Data Availability Statement:** All relevant data are within the manuscript and its Supporting Information files.

**Funding:** The CNPq - Conselho Nacional de Desenvolvimento Científico e Tecnológico - Brasil had the role of providing the study grant for MTM (proc. nˆ103587/2019-5); and PETROBRAS (Grant Number 2018/00545-0) had the role of paying both the publication fee and the professional language editing service. The funders had no role in study design, data collection and analysis, decision to publish, or preparation of the manuscript.

**Competing interests:** PETROBRAS (Grant Number 2018/00545-0) had the role of paying both the publication fee and the professional language editing service. This does not alter our adherence to PLOS ONE policies on sharing data and materials.

accordingly, by increasing the redundancy of the system; and to mitigate (or even estimate) a possible systematic effect [1, 2].

The models are usually formulated with minimal constraint or extra (redundant) constraints. In that case, we refer to the so-called *equality constraints*, which are usually incorporated into a system of equations to create a well-posed model [3]. For the most part, minimal constraints are introduced to solve to the problem of rank deficiency in linear (or linearized) systems. The rank deficiency is often caused by the lack (or insufficient) information about a problem. In the field of geodesy, for example, minimal constraints are external information whose primary role is to specify the coordinate system to which the network station positions will be estimated by the least-squares method (LS). This problem is known as *datum definition* (or also zero-order design or datum choice problem) [4–9]. Several works have investigated the minimum-constrained adjustment and the datum choice problem in the geodetic literature, focusing on topics like free-adjustment and the role of inner constraints [10–13].

If the number of constraints exceeds the minimum needed to solve the rank deficiency of the equation systems, we say that we have redundant (or extra) constraints. Extra constraints are also used to check the stability of points in geodetic deformation analysis [14–16] to test the compatibility of constraints with the observations and the rest of the constraints [17–19].

So far we have only distinguished the constraints in terms of numerical quantity. The model can also be subject to a *hard* and *soft* (or *weighted*) constraints. Hard constraints can often represent a case in which there exist known functional relations between the unknown parameters. Soft constraints (or *looser* constraints) are, however, for when functional relations can be slightly violated depending on their uncertainty [2, 19]. Soft constraints may also be referred to as a pseudo-observation model [20].

The well-known least-squares (LS) is widely used as a standard method of estimating model parameters in geodetic applications and many others branches of modern science [21–41]. This is due to the flexibility of the LS, since no concepts from probability theory are used in formulating the least-squares minimization problem.

LS is a linear unbiased estimator (LUE), and in some special cases, it coincides with the best linear unbiased estimator (BLUE). The estimator that has the smallest variance of all LUEs is called the best linear unbiased estimator (BLUE). If we have full knowledge of the probability density function (PDF) of the measurements, the method of maximum likelihood estimation (MLE) can also be applied. In case of normally distributed measurements (Gauss–Markov model), the MLE estimators are identical to the BLUE ones, and therefore the LS and MLE principles provide identical results [24, 42]; however, the presence of undesirable outliers in the dataset makes LS no longer *unbiased* and not coincide with MLE [43].

Here, we assume that an outlier is an observation that has deviated from its most probable value to the point of jeopardizing the mathematical model (functional and stochastic) to which it should belong. Due to its importance, outliers must be appropriately treated to ensure the quality of data analysis [44–50].

In this study, we employed iterative data snooping (*IDS*), which is a hypothesis test-based outlier. It is important to mention that *IDS* is not restricted to the field of geodetic statistics, but is a generally applicable method [51, 52]. *IDS* is an iterative outlier elimination procedure, which combines estimation, testing and a corrective action [44, 53]. Parameter estimation is often conducted using LS. Then, hypothesis testing is performed with the aim to identify any outlier that may be present in the dataset. After identification, the suspected outlier is then excluded from the dataset as a corrective action (i.e., adaptation), and the LS is restarted without the rejected measurement. If the model redundancy permits, this procedure is repeated

until no more (possible) outliers can be identified (see e.g., [23], pp. 135). Although in this study, we restricted ourselves to the case of one outlier at a time, *IDS* can also be applied for cases containing multiple (simultaneous) outliers [54]. For more details about multiple (simultaneous) outliers, the reader is referred to [55–57]. Because *ÌDS* is based on statistical hypothesis testing, there are chances of both correct and incorrect decisions. Recently, Rofatto et al. [44] provided an algorithm based on Monte Carlo to determine the probability levels associated with *IDS*. In that case, they described six classes of decisions for *IDS*, namely probability of correct identification ($\mathcal{P}_{CI}$), probability of missed detection ($\mathcal{P}_{MD}$), probability of wrong exclusion ($\mathcal{P}_{WE}$), probability of over-identification positive ($\mathcal{P}_{over+}$), probability of over-identification negative ($\mathcal{P}_{over-}$) and statistical overlap ($\mathcal{P}_{ol}$), as follows:

- $\mathcal{P}_{CI}$: Probability of identifying and removing correctly an outlying measurement;

- $\mathcal{P}_{MD}$: Probability of not detecting the outlier (i.e., Type II decision error for *IDS*);

- $\mathcal{P}_{WE}$: Probability of identifying and removing a non-outlying measurement while the '*true*' outlier remains in the dataset (i.e., Type III decision error [58] for *IDS*);

- $\mathcal{P}_{over+}$: Probability of identifying and removing correctly the outlying measurement and others;

- $\mathcal{P}_{over-}$: Probability of identifying and removing more than one non-outlying measurement, whereas the 'true outlier' remains in the dataset;

- $\mathcal{P}_{ol}$: occurs in cases where one alternative hypothesis has the same distribution as the another one. These hypotheses cannot be distinguished because their test statistics are numerically the same, violating the *IDS* rule of one outlier at a time. In that case, they are non-separable and an outlier cannot be identified. In other words, it corresponds to the probability of flagging simultaneously two (or more) measurements as outliers.

Based on the probabilities of correct detection ($\mathcal{P}_{CD} = 1 - \mathcal{P}_{MD}$) and correct identification ($\mathcal{P}_{CI}$), the minimal biases, MDB (minimal detectable bias) and MIB (minimal identifiable bias), can be computed as sensitivity indicators for outlier detection and identification, respectively. "Outlier Detection" only informs whether or not there might have been at least one outlier; however, the detection does not tell us which measurement is an outlier. The localization of the outlier is a problem of "outlier identification", i.e., "Outlier Identification" implies the execution of a search among the measurements for the most likely outlier [44]; therefore, the smallest value of an outlier that can be detected, given a certain $\mathcal{P}_{CD}$, defines the MDB. On the other hand, the smallest value of an outlier that can be identified, given a certain $\mathcal{P}_{CI}$, defines the MIB.

In this study, we investigated the effects of models subject to constraints (minimum, redundant, hard and soft) on the probability levels associated with *IDS*. It is important to emphasize that if a standard deviation of a constraint (or a set of a constraint) is changed from zero to a non-zero value, it is called a "relaxation" of the constraint [20].

We also evaluated the effect of relaxing constraints on the MIB and MDB. This kind of assessment is a kind of sensitivity analysis. We also highlight that the task of clustering a set of geodetic measurements was applied for the first time in this study. We intend to show that the clusters can be defined according to two deterministic parameters: local redundancy and correlation between the outlier test statistics.

Critical values optimized by the Monte Carlo method were used here [44, 51] in order to compute the decision classes associated with *IDS*, i.e., $\mathcal{P}_{CI}$, $\mathcal{P}_{MD}$, $\mathcal{P}_{WE}$, $\mathcal{P}_{over+}$, $\mathcal{P}_{over-}$ and $\mathcal{P}_{ol}$.

## Material and methods

We used the procedure provided by Rofatto et al. [44] to compute the probability levels associated with *IDS*, as well as to estimate the both MDB and MIB. The procedure is summarized in Fig 1.

The probability levels associated with *IDS* were computed for each observation individually and for each outlier magnitude; however, they were grouped into clusters based on number of local redundancy ($r_i$) and maximum absolute correlation between the outlier test statistics ($\rho_{w_i,w_j}$). Furthermore, we took care to control the *family-wise error rate*. See Supporting Information for more details S1 Appendix.

## Problem description

To analyze the effects of the constraints on the *IDS*, an example was taken from a geodetic leveling network with 12 height differences between the points. The equipment used to measure the level difference was an electronic digital level. In that case, the leveling measurement system comprises of a special bar-coded staff (also called barcode rod) and a digital level (instrument). A digital level is basically a telescope that enables a horizontal line of sight. Digital levels consist of additional electronic image processing components to automatically read and analyze digital (bar coded) leveling staffs, where the graduation is replaced by a manufacturer dependent code pattern. Generally, the result is automatically stored in the data collector of the digital level. An example of a "*digital level—bar-code staff*" system is displayed in the Fig 2. For more details about digital level see e.g., [59–61].

The standard deviation of the uncorrelated measurements were the same and taken equal to $\sigma = 1mm$. The points are indicated as A to G. The eight network configuration are displayed in Fig 3a–3e and their details are given as follows:

1. Fig 3a: Network with 1 hard constraint (i.e., network minimally constrained). Since the dimension of the network is 1D, the minimum information necessary to estimate the unknown heights is one. The height of G was fixed as a control point (hard constraint), and 6 unknown heights (A,B,C,D,E,F) were minimally constrained; therefore, the redundancy of the system (or overall degrees of freedom) was $r = n\text{-}rank(A) = n - u = 12 - 6 = 6$.

2. Fig 3b: Network with 1 extra hard constraint (i.e., two hard constraints). The heights A and D were taken as hard constraints (i.e., heights A and D were fixed). The redundancy of the system in that case was $r = 12 - 5 = 7$ with 5 unknown heights (B,C,E,F,G) over-constrained.

3. Fig 3c: Network with 2 extra hard constraints (i.e., three hard constraints). The heights A, D and G were taken as hard constraints. In that case, the redundancy of the system was $r = 12 - 4 = 8$.

4. Fig 3d: Network with 2 soft constraints (A and D). In that case, a standard deviation larger than zero was assigned to both constraints i.e., $\sigma_c > 0$. In other words, A and D were processed as being both observations and unknown parameters, i.e., A and D were pseudo-observations. Both constraints were simultaneously *relaxed* by considering their uncertainties 10 times worse than the measurements (i.e., $\sigma_c = 10 \times \sigma = 10mm$); 10 times better than measurements (i.e., $\sigma_c = 0.1mm$); their uncertainties equal to the measurements ($\sigma_c = 1mm$). In that case, the redundancy of the system was $r = 14 - 7 = 7$.

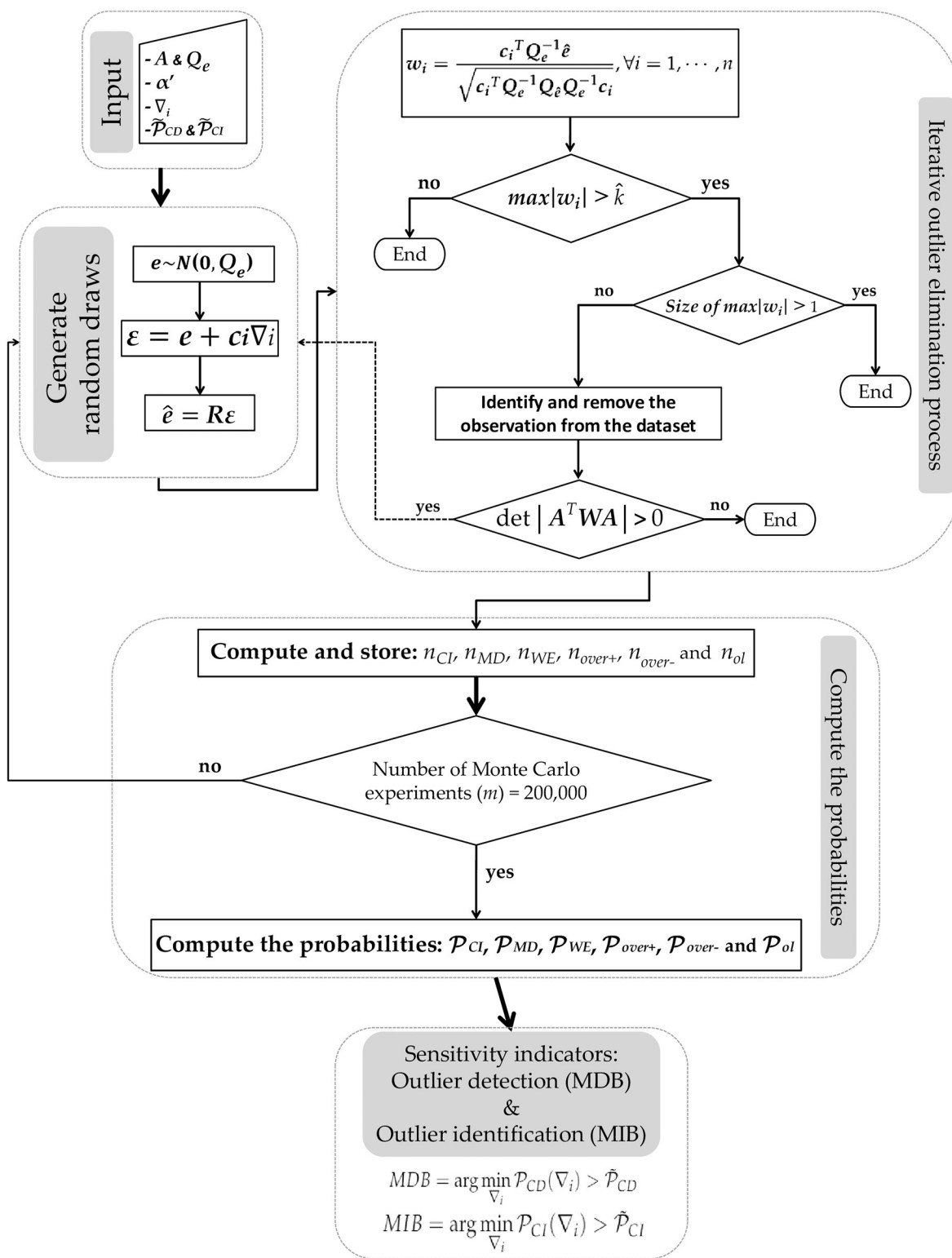

**Fig 1. Flowchart of the algorithm.** Flowchart of the algorithm to compute the probability levels of Iterative Data Snooping (IDS) for each measurement in the presence of an outlier [44].

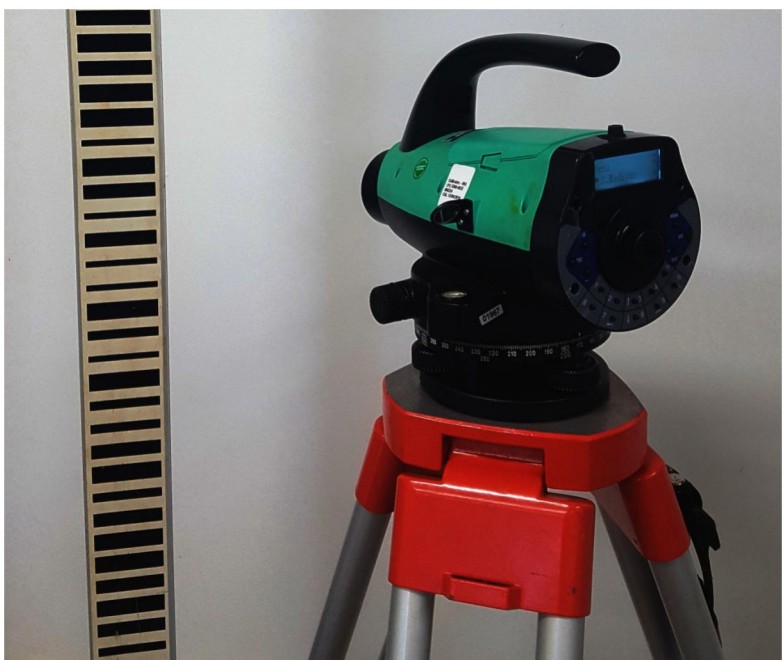

**Fig 2. Digital level—Bar-code staff system.** Example of a *digital level—bar-code staff* system [44].

5. [Fig 3e](): Network processed with A, D and G as pseudo-observations. Those three constraints were simultaneously *relaxed* by considering their standard deviations equal to $\sigma_c = 10mm$ (10 times worse than measurements); $\sigma_c = 0.1mm$ (10 times better than measurements); $\sigma_c = 1mm$ (the same as the measurements). In that case, the redundancy of the system was $r = 15 - 7 = 8$.

The following system of equations for that problem is given by:

$$y_1 + e_1 \quad = h_B - h_A$$

$$y_2 + e_2 \quad = h_C - h_B$$

$$y_3 + e_3 \quad = h_D - h_C$$

$$\vdots$$

$$y_7 + e_7 \quad = h_B - h_G \qquad (1)$$

$$y_8 + e_8 \quad = h_C - h_G$$

$$\vdots$$

$$y_{11} + e_{11} \quad = h_B - h_F$$

$$y_{12} + e_{12} \quad = h_C - h_E$$

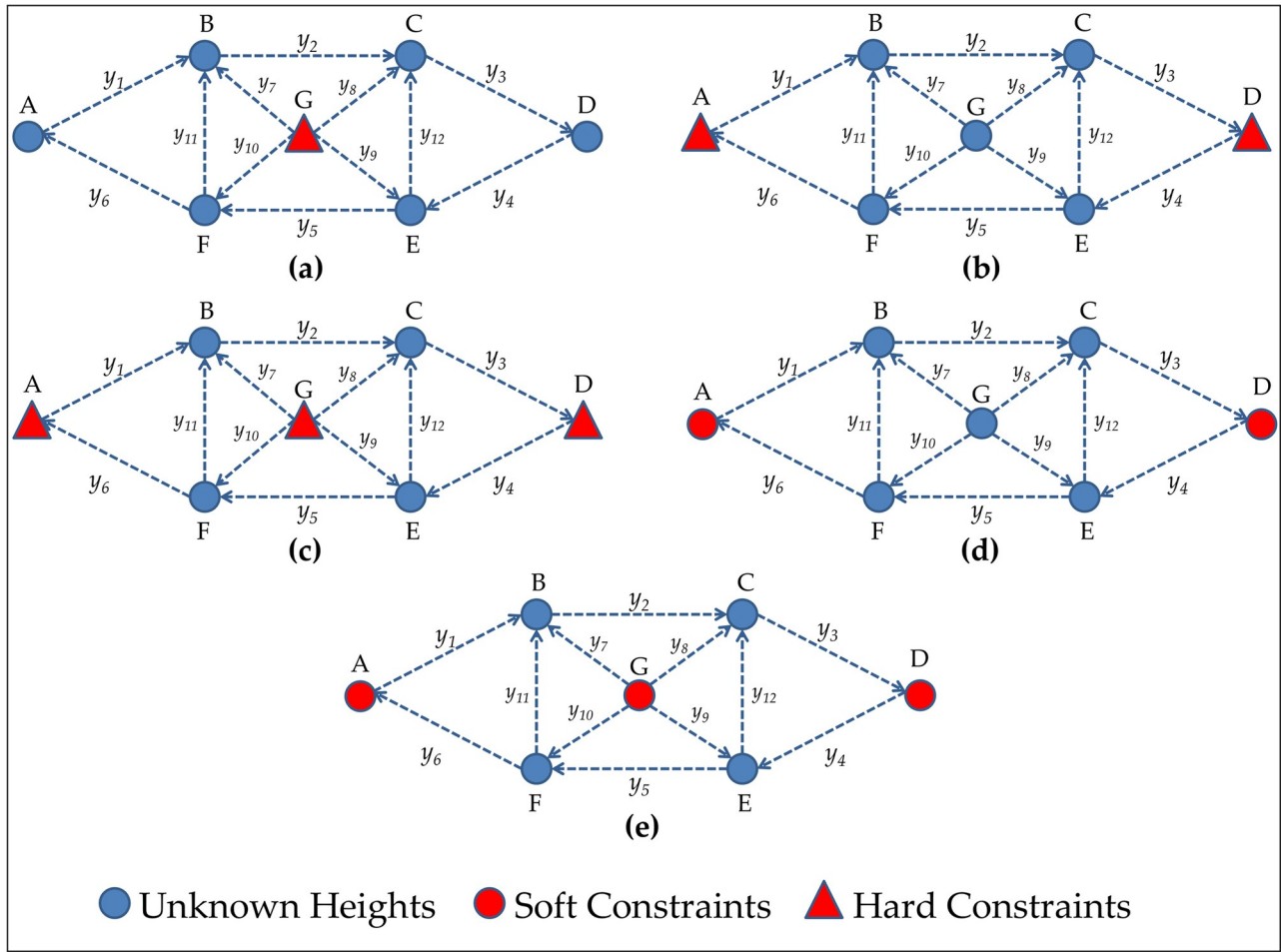

**Fig 3. Different constraint scenarios.** Leveling geodetic network subject to different constraint scenarios.

The design matrix ($A$) for the system of equations in 1 is given by:

$$A = \begin{bmatrix} -1 & 1 & 0 & 0 & 0 & 0 & 0 \\ 0 & -1 & 1 & 0 & 0 & 0 & 0 \\ 0 & 0 & -1 & 1 & 0 & 0 & 0 \\ 0 & 0 & 0 & -1 & 1 & 0 & 0 \\ 0 & 0 & 0 & 0 & -1 & 1 & 0 \\ 1 & 0 & 0 & 0 & 0 & -1 & 0 \\ 0 & 1 & 0 & 0 & 0 & 0 & -1 \\ 0 & 0 & 1 & 0 & 0 & 0 & -1 \\ 0 & 0 & 0 & 0 & 1 & 0 & -1 \\ 0 & 0 & 0 & 0 & 0 & 1 & -1 \\ 0 & 1 & 0 & 0 & 0 & -1 & 0 \\ 0 & 0 & 1 & 0 & -1 & 0 & 0 \end{bmatrix} \tag{2}$$

Note that the rank defect of the matrix $A$ is $u\text{-}rank(A) = 7 - 6 = 1$. In that case, at least one constraint is needed in order to avoid rank the deficiency of the matrix $A$. This is guaranteed when one height is known. For example, from the network in Fig 3a, we have added the height

G as known (i.e., as a hard constraint). In that case, the constraint equation should be added into the system in 1, i.e.,

$$y_{13} = h_G \ with \ \sigma_{y_{13}} = 0, \tag{3}$$

noticing that because the standard deviation is zero, the observation is non-stochastic (hard constraint) and the residual $e_{y_{13}} = 0$. This can generate problems in the inversion of the covariance matrix of the observations $Q_e$ for the calculation of the weight matrix $W$, because the weight for that constraint would be undefined, i.e., $\frac{1}{0}$. In order to avoid that problem, we have eliminated the rank deficiency of matrix $A$ by removing the seventh column of matrix $A$ in 2 associated with the height G. Now, we have $u$-$rank(A) = 6 - 6 = 0$. The constraint defines the geodetic datum, i.e., the S-system [62]. Another approach to solving the system of equations in 1 could be based on generalized (pseudo) inverses [63].

The location of the constraints can be chosen in some circumstances, for example, during the design stage of a geodetic network. For the special case of having a minimally constrained system, the location of the constraint will not influence the $w$-test statistics and the sensitivity indicators (MIB and MDB) [9]; however, more constraints than the minimum necessary to have a solution (i.e., extra constraints or redundant constraints) can change the least-squares residuals and hence $w$-test statistics and the minimal biases.

From the network with one extra constraint (2 constraints) in Fig 3b, for example, both the first (height A) and fourth column (height D) of matrix $A$ in 2 were eliminated in the case of having the two heights as hard constraints. For the case where these two heights (A and D) were taken as soft constraints, however, two observation equations were added to Eq 1, i.e.,

$$\begin{aligned} y_{13} + e_{13} &= h_A, \ \sigma_{y_{13}} > 0 \\ y_{14} + e_{14} &= h_D, \ \sigma_{y_{14}} > 0 \end{aligned} \tag{4}$$

In the case of soft constraints in Eq 4, 2 lines were added in matrix $A$. In other words, A and D were taken as pseudo-observations. In that case, the rank deficiency was also null (i.e., $u$-$rank(A) = 7 - 7 = 0$), the redundancy of the system was $r = n$-$rank(A) = n - u = 7$ and the matrix $A$ was given as follows:

$$A = \begin{bmatrix} -1 & 1 & 0 & 0 & 0 & 0 & 0 \\ 0 & -1 & 1 & 0 & 0 & 0 & 0 \\ 0 & 0 & -1 & 1 & 0 & 0 & 0 \\ 0 & 0 & 0 & -1 & 1 & 0 & 0 \\ 0 & 0 & 0 & 0 & -1 & 1 & 0 \\ 1 & 0 & 0 & 0 & 0 & -1 & 0 \\ 0 & 1 & 0 & 0 & 0 & 0 & -1 \\ 0 & 0 & 1 & 0 & 0 & 0 & -1 \\ 0 & 0 & 0 & 0 & 1 & 0 & -1 \\ 0 & 0 & 0 & 0 & 0 & 1 & -1 \\ 0 & 1 & 0 & 0 & 0 & -1 & 0 \\ 0 & 0 & 1 & 0 & -1 & 0 & 0 \\ \mathbf{1} & 0 & 0 & 0 & 0 & 0 & 0 \\ 0 & 0 & 0 & \mathbf{1} & 0 & 0 & 0 \end{bmatrix} \tag{5}$$

For this example of 2 soft constraints, and by considering the both soft constraints with standard deviation $\sigma_c = 10mm$, the symmetric and positive semi-definite covariance matrix of the observations ($\boldsymbol{Q}_e$) was given as follows:

$$\boldsymbol{Q}_e = \begin{bmatrix} 1 & 0 & 0 & \cdots & 0 & 0 \\ 0 & 1 & 0 & \cdots & 0 & 0 \\ 0 & 0 & 1 & \cdots & 0 & 0 \\ \vdots & \vdots & \vdots & \ddots & \vdots & \vdots \\ 0 & 0 & 0 & \cdots & 100 & 0 \\ 0 & 0 & 0 & \cdots & 0 & 100 \end{bmatrix} \tag{6}$$

The last two rows and columns of the matrix $\boldsymbol{Q}_e$ in 6 refer to the variances $(\sigma_c^2 = (10mm)^2 = 100mm^2)$ of the heights constraints A and D, respectively. Similarly, matrices $\boldsymbol{A}$ and $\boldsymbol{Q}_e$ were constructed for the other cases studied here.

Although the measurements are able to identify an outlier for the case of having only one single soft constraint, the pseudo-observation (constraint) is not. In that case, the defect configuration is associated with the additional parameter in the constraint (i.e., the presence of an outlier in the constraint). In other words, an additional parameter on the soft constraint will not estimable. For example, if the height point G was taken as a soft constraint, the presence of an outlier in pseudo-observation G would lead to rank deficiency of matrix $\boldsymbol{A}$, i.e., $u\text{-}rank(A) = 8 - 7 = 1$; therefore, the case of having only one single soft constraint was not considered here.

## Result of the hard constraint effects on the iterative outlier elimination procedure

The scenarios in Fig 3a (network minimally constrained), Fig 3b (two hard constraints) and Fig 3c (three hard constraints) were considered here for the analysis. Table 1 gives the local

**Table 1. Local redundancy ($r_i$), standard deviation of the least-squares (LS)-estimated outlier $\sigma_{\nabla_i}$ and the maximum absolute correlation ($max_{\rho_{w_i,w_j}}$) for each scenario of hard constraint.**

| Measurement | 1 hard constraint | | | 2 hard constraints | | | 3 hard constraints | | |
|---|---|---|---|---|---|---|---|---|---|
| | $r_i$ | $\sigma_{\nabla_i}$ | $max_{\rho_{w_i,w_j}}$ | $r_i$ | $\sigma_{\nabla_i}$ | $max_{\rho_{w_i,w_j}}$ | $r_i$ | $\sigma_{\nabla_i}$ | $max_{\rho_{w_i,w_j}}$ |
| $y_1$ | 0.396 | 1.589 | 1.00 | 0.583 | 1.309 | 0.36 | 0.708 | 1.188 | 0.41 |
| $y_2$ | 0.500 | 1.414 | 0.47 | 0.583 | 1.309 | 0.36 | 0.583 | 1.309 | 0.32 |
| $y_3$ | 0.396 | 1.589 | 1.00 | 0.583 | 1.309 | 0.36 | 0.708 | 1.188 | 0.41 |
| $y_4$ | 0.396 | 1.589 | 1.00 | 0.583 | 1.309 | 0.36 | 0.708 | 1.188 | 0.41 |
| $y_5$ | 0.500 | 1.414 | 0.47 | 0.583 | 1.309 | 0.36 | 0.583 | 1.309 | 0.32 |
| $y_6$ | 0.396 | 1.589 | 1.00 | 0.583 | 1.309 | 0.36 | 0.708 | 1.188 | 0.41 |
| $y_7$ | 0.563 | 1.333 | 0.47 | 0.583 | 1.309 | 0.36 | 0.708 | 1.188 | 0.41 |
| $y_8$ | 0.563 | 1.333 | 0.47 | 0.583 | 1.309 | 0.36 | 0.708 | 1.188 | 0.41 |
| $y_9$ | 0.563 | 1.333 | 0.47 | 0.583 | 1.309 | 0.36 | 0.708 | 1.188 | 0.41 |
| $y_{10}$ | 0.563 | 1.333 | 0.47 | 0.583 | 1.309 | 0.36 | 0.708 | 1.188 | 0.41 |
| $y_{11}$ | 0.583 | 1.309 | 0.43 | 0.583 | 1.309 | 0.36 | 0.583 | 1.309 | 0.32 |
| $y_{12}$ | 0.583 | 1.309 | 0.43 | 0.583 | 1.309 | 0.36 | 0.583 | 1.309 | 0.32 |

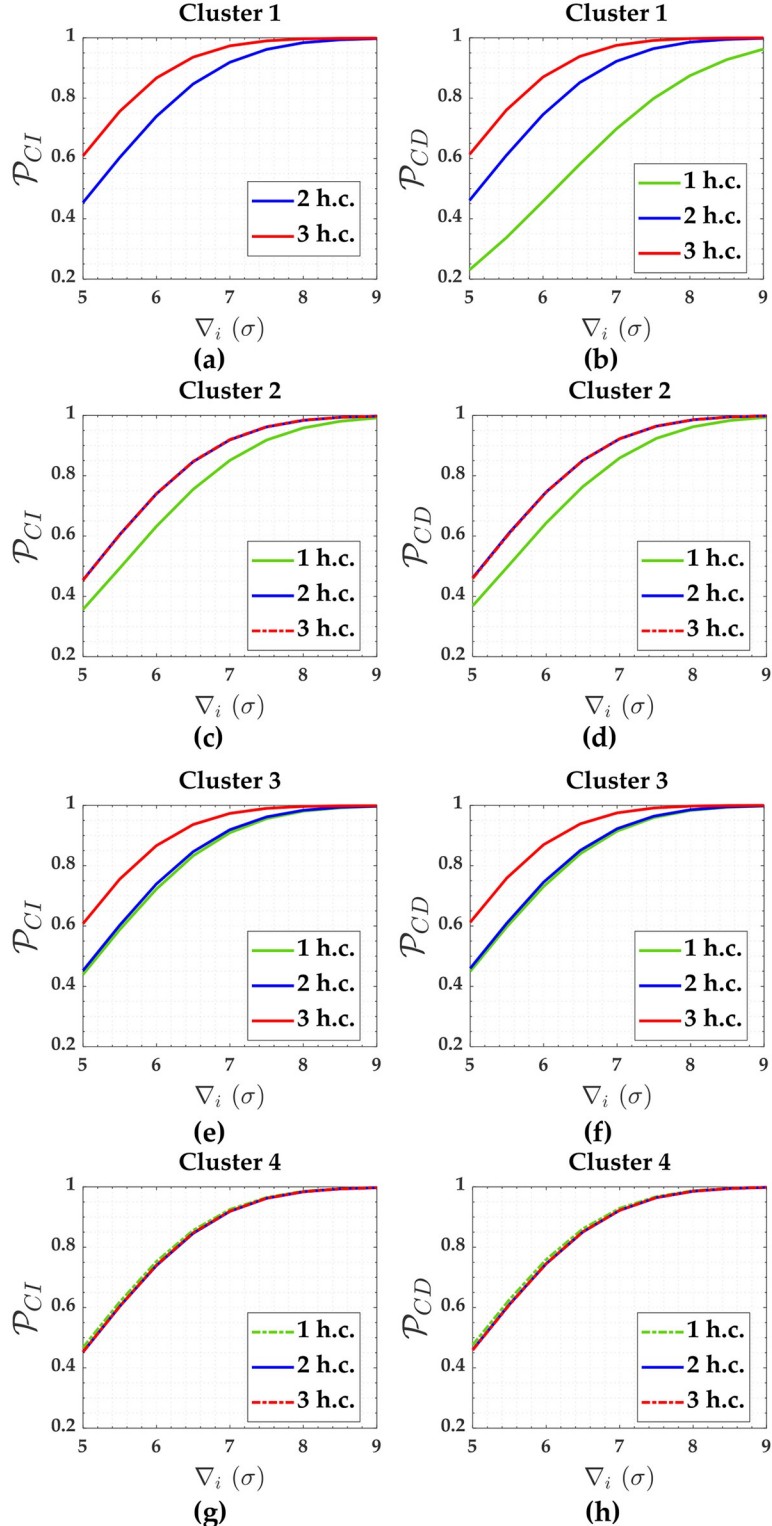

**Fig 4.** $\mathcal{P}_{CI}$ and $\mathcal{P}_{CD}$ **for the case of hard constraints and for $\alpha' = 0.001$.** Cluster 1(**A,b**), Cluster 2(**c,d**), Cluster 3(**e,f**) and Cluster 4(**g,h**).

**Table 2. MDB (minimal detectable bias) and MIB (minimal identifiable bias) for the case of hard constraints based on $\alpha' = 0.001$ and $\tilde{\mathcal{P}}_{CD} = \tilde{\mathcal{P}}_{CI} = 0.8$.**

| Cluster | 1 hard constraint | | 2 hard constraints | | 3 hard constraints | |
|---|---|---|---|---|---|---|
| | MDB ($\sigma$) | MIB ($\sigma$) | MDB ($\sigma$) | MIB ($\sigma$) | MDB ($\sigma$) | MIB ($\sigma$) |
| 1 | 7.5 | - | 6.3 | 6.3 | 5.7 | 5.7 |
| 2 | 6.7 | 6.8 | 6.3 | 6.4 | 6.3 | 6.4 |
| 3 | 6.4 | 6.4 | 6.3 | 6.3 | 5.8 | 5.8 |
| 4 | 6.4 | 6.4 | 6.4 | 6.4 | 6.4 | 6.4 |

redundancy ($r_i$), the standard deviation of the LS-estimated outlier $\sigma_{\nabla_i}$ and the maximum absolute correlation ($max_{\rho_{w_i,w_j}}$) for each scenario of hard constraint set out in this study, i.e., Fig 3a–3c.

Next, the twelve leveling measurements were clustered into four clusters. The four cluster were defined as follows:

- Cluster 1: $y_1$, $y_3$, $y_4$ and $y_6$.

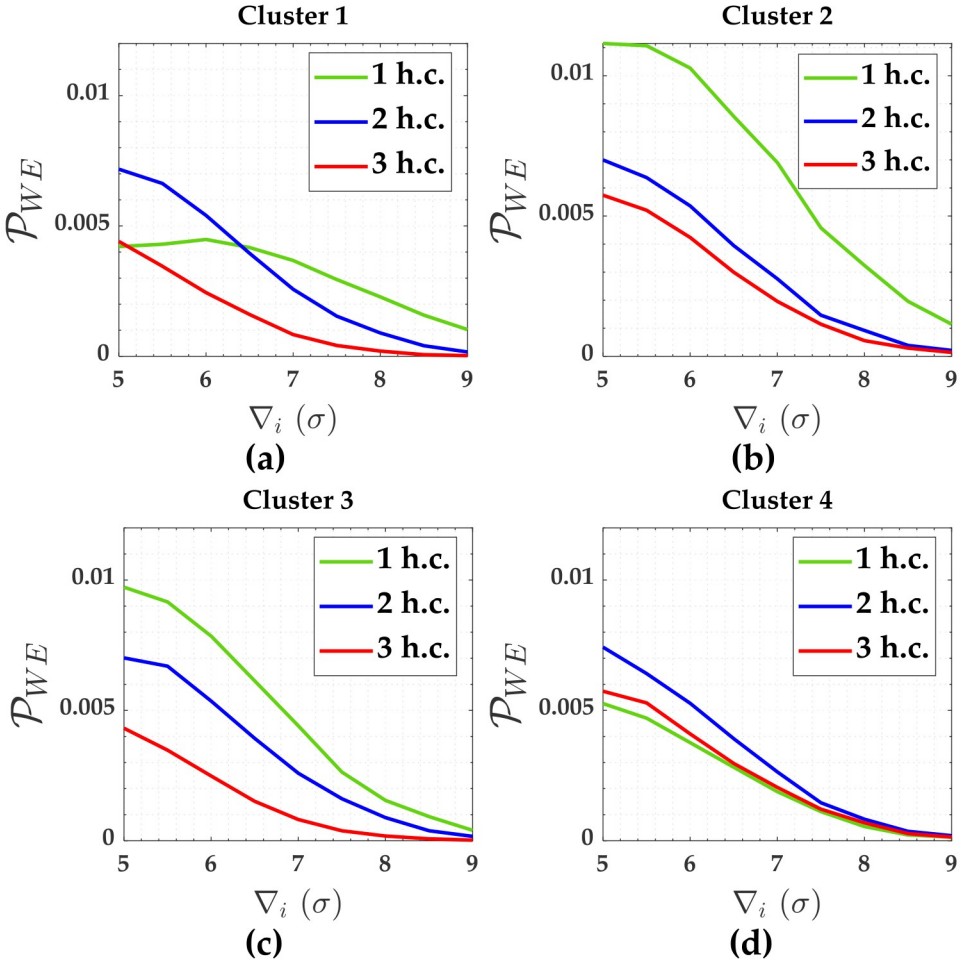

**Fig 5. $\mathcal{P}_{WE}$ for the case of hard constraints and for $\alpha' = 0.001$.** Cluster 1(**A**), Cluster 2(**b**), Cluster 3(**c**) and Cluster 4 (**d**).

- Cluster 2: $y_2$ and $y_5$.

- Cluster 3: $y_7$, $y_8$, $y_9$ and $y_{10}$.

- Cluster 4: $y_{11}$ and $y_{12}$.

The probability levels associated with *IDS* were averaged for each of these clusters. The critical values were $\hat{k} = 3.89$, $\hat{k} = 3.93$ and $\hat{k} = 3.93$ for one hard constraint, two hard constraints and three hard constraints, respectively. These critical values were found for $\alpha' = 0.001$. $\mathcal{P}_{CI}$ and $\mathcal{P}_{CD}$ and are displayed in Fig 4 for each number of hard constraint (denoted by **h.c.**).

The outlier magnitude were defined from $|5\sigma|$ to $|9\sigma|$. The outlier of $|5\sigma|$ was chosen because it is approximately the lowest $MDB_{0(i)}$ of the network when a single hypothesis testing is in play (See Supplementary Material for more details S1 Appendix). That $MDB_{0(i)}$ of $|5\sigma|$ was computed for a significance level of $\alpha' = 0.001$ and a power of the test $\gamma_0 = 0.8$. This strategy reduces the search space for an MIB, because we will always have the following inequality $MIB \geq MDB_{0(i)}$ [52, 64]. Remember that the *IDS* procedure is an example of multiple hypothesis testing. The success rate for outlier detection and outlier identification were taken as being $\tilde{\mathcal{P}}_{CD} = \tilde{\mathcal{P}}_{CI} = 0.8$, respectively. Table 2 provides the values of MDB and MIB for that case of hard constraints.

Fig 5 shows the $\mathcal{P}_{WE}$. $\mathcal{P}_{over+}$ and $\mathcal{P}_{over-}$ were smaller than 0.001 (i.e., they were practically null). There were not $\mathcal{P}_{ol}$ for clusters 2, 3 and 4. We will discuss more about $\mathcal{P}_{ol}$ later.

## Result of the soft constraint effects on the iterative outlier elimination procedure

Both configurations in Fig 3d and 3e were analyzed in terms of soft constraints. In that case, the critical values were $\hat{k} = 3.95$, $\hat{k} = 3.95$ and $\hat{k} = 3.92$ for two soft constraints with $\sigma_c = 0.1mm$, $\sigma_c = 1mm$ and $\sigma_c = 10mm$, respectively. In the case of three soft constraints, the critical values found were $\hat{k} = 3.99$, $\hat{k} = 3.99$ and $\hat{k} = 3.96$ for $\sigma_c = 0.1mm$, $\sigma_c = 1mm$ and $\sigma_c = 10mm$, respectively. All these critical values were computed for $\alpha' = 0.001$. Table 3 gives the local

**Table 3. Local redundancy ($r_i$), standard deviation of the LS-estimated outlier $\sigma_{\nabla_i}$ (mm) and the maximum absolute correlation ($max_{\rho_{w_i . w_j}}$) for each scenario of two soft constraints.**

| Measurement | $\sigma_c = 0.1mm$ | | | $\sigma_c = 1mm$ | | | $\sigma_c = 10mm$ | | |
|---|---|---|---|---|---|---|---|---|---|
| | $r_i$ | $\sigma_{\nabla_i}$ | $max_{\rho_{w_i . w_j}}$ | $r_i$ | $\sigma_{\nabla_i}$ | $max_{\rho_{w_i . w_j}}$ | $r_i$ | $\sigma_{\nabla_i}$ | $max_{\rho_{w_i . w_j}}$ |
| $y_1$ | 0.581 | 1.312 | 0.564 | 0.471 | 1.457 | 0.681 | 0.397 | 1.587 | 0.994 |
| $y_2$ | 0.582 | 1.311 | 0.376 | 0.533 | 1.369 | 0.423 | 0.501 | 1.413 | 0.471 |
| $y_3$ | 0.581 | 1.312 | 0.564 | 0.471 | 1.457 | 0.681 | 0.397 | 1.587 | 0.994 |
| $y_4$ | 0.581 | 1.312 | 0.564 | 0.471 | 1.457 | 0.681 | 0.397 | 1.587 | 0.994 |
| $y_5$ | 0.582 | 1.311 | 0.376 | 0.533 | 1.369 | 0.423 | 0.501 | 1.413 | 0.471 |
| $y_6$ | 0.581 | 1.312 | 0.564 | 0.471 | 1.457 | 0.681 | 0.397 | 1.587 | 0.994 |
| $y_7$ | 0.583 | 1.310 | 0.359 | 0.571 | 1.324 | 0.423 | 0.563 | 1.333 | 0.471 |
| $y_8$ | 0.583 | 1.310 | 0.359 | 0.571 | 1.324 | 0.423 | 0.563 | 1.333 | 0.471 |
| $y_9$ | 0.583 | 1.310 | 0.359 | 0.571 | 1.324 | 0.423 | 0.563 | 1.333 | 0.471 |
| $y_{10}$ | 0.583 | 1.310 | 0.359 | 0.571 | 1.324 | 0.423 | 0.563 | 1.333 | 0.471 |
| $y_{11}$ | 0.583 | 1.309 | 0.358 | 0.583 | 1.309 | 0.398 | 0.583 | 1.309 | 0.433 |
| $y_{12}$ | 0.583 | 1.309 | 0.358 | 0.583 | 1.309 | 0.398 | 0.583 | 1.309 | 0.433 |
| $y_{13}$ | 0.007 | 1.163 | 1.000 | 0.300 | 1.826 | 1.000 | 0.497 | 14.189 | 1.000 |
| $y_{14}$ | 0.007 | 1.163 | 1.000 | 0.300 | 1.826 | 1.000 | 0.497 | 14.189 | 1.000 |

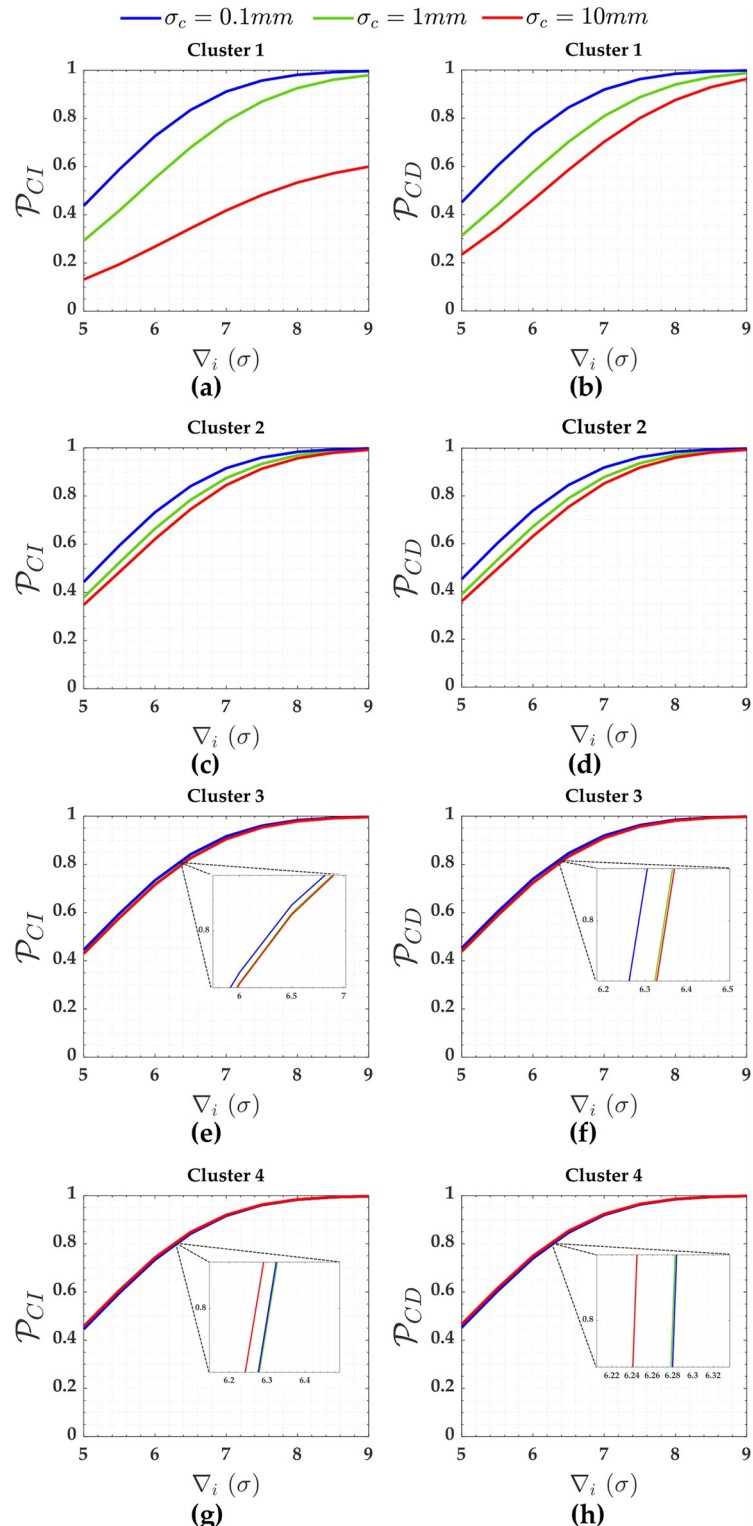

**Fig 6. $\mathcal{P}_{CI}$ and $\mathcal{P}_{CD}$ for the measurements subject to the scenarios of two soft constraints for $\alpha' = 0.001$.** Cluster 1(**a**, **b**), Cluster 2(**c**,**d**), Cluster 3(**e**,**f**) and Cluster 4(**g**,**h**).

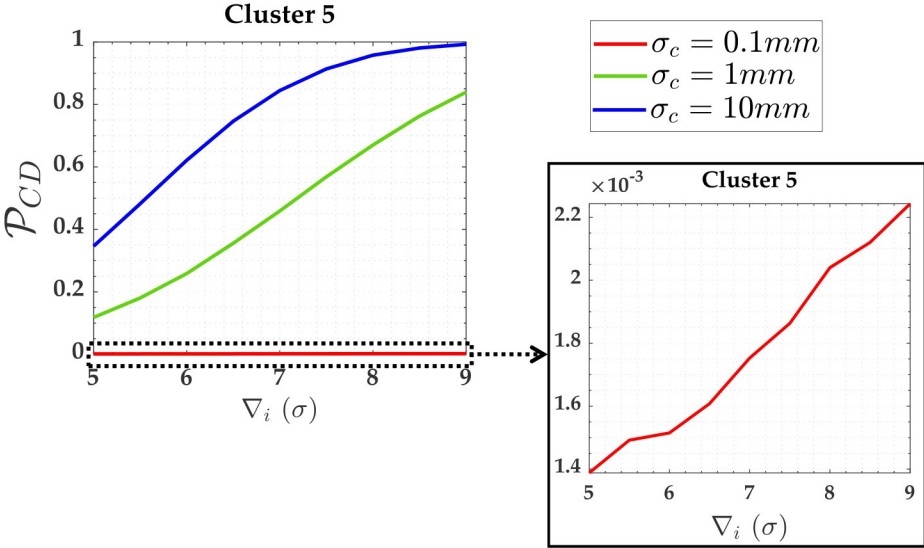

**Fig 7. Probability of $\mathcal{P}_{CD}$ and $\mathcal{P}_{CI}$ for the two soft constraints and for $\alpha' = 0.001$.** Cluster 5: heights A and D.

redundancy ($r_i$), the standard deviation of the LS-estimated outlier $\sigma_{\nabla_i}$ and the maximum absolute correlation ($max_{\rho_{w_i.w_j}}$) for the scenarios of two constraints.

From Table 3, five clusters were defined for each case of two soft constraints, i.e., for the case where heights A and D were given as soft constraints in Fig 3d, as follows:

- Cluster 1: $y_1$, $y_3$, $y_4$ and $y_6$.

- Cluster 2: $y_2$ and $y_5$.

- Cluster 3: $y_7$, $y_8$, $y_9$ and $y_{10}$.

- Cluster 4: $y_{11}$ and $y_{12}$.

- Cluster 5: $y_{13}$ and $y_{14}$.

$\mathcal{P}_{CI}$ and $\mathcal{P}_{CD}$ for the measurements (Cluster 1 to Cluster 4) subject to the scenarios of two soft constraints (heights A and D) are displayed in Fig 6.

Note that Cluster 5 is associated with the two soft constraints (i.e., $y_{13}$ and $y_{14}$). The $\mathcal{P}_{CI}$ for these both soft constraints were null; however, $\mathcal{P}_{CD}$ were not. Fig 7 shows $\mathcal{P}_{CD}$ for these two soft constraints (i.e., heights A and D).

The $\mathcal{P}_{WE}$ for the measurements (Cluster 1 to Cluster 4) subject to the scenarios of two soft constraints (heights A and D) are displayed in Fig 8. Fig 9 gives $\mathcal{P}_{WE}$ for two constraints (i.e., heights A and D). The $\mathcal{P}_{over+}$ and $\mathcal{P}_{over-}$ and the $\mathcal{P}_{ol}$ were practically null for that case. The sensitivity indicators (MDB and MIB) for each scenario of two soft constraints are displayed in Table 4.

Table 5 gives the local redundancy ($r_i$), the standard deviation of the LS-estimated outlier $\sigma_{\nabla_i}$ and the maximum absolute correlation ($max_{\rho_{w_i.w_j}}$) for the scenarios of three soft constraints.

The $\mathcal{P}_{CI}$ and $\mathcal{P}_{CD}$ in Fig 10 were computed for the clusters based on Table 5, as follows:

- Cluster 1: $y_1$, $y_3$, $y_4$ and $y_6$.

- Cluster 2: $y_2$ and $y_5$.

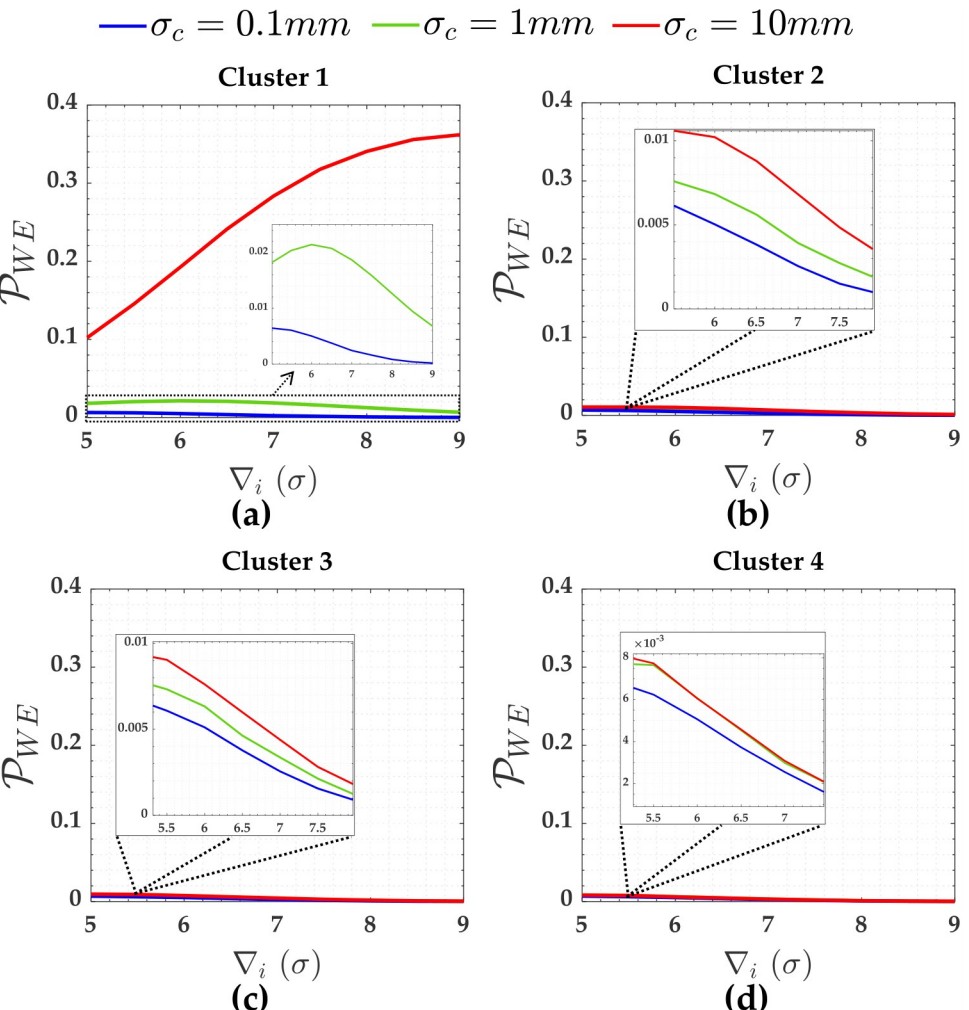

**Fig 8. The $\mathcal{P}_{WE}$ for the measurements subject to the scenarios of two soft constraints for $\alpha' = 0.001$.** Cluster 1(**a**), Cluster 2(**b**), Cluster 3(**c**) and Cluster 4(**d**).

- Cluster 3: $y_7$, $y_8$, $y_9$ and $y_{10}$.

- Cluster 4: $y_{11}$ and $y_{12}$.

- Cluster 5: $y_{13}$ and $y_{14}$.

- Cluster 6: $y_{15}$.

Fig 11 shows $\mathcal{P}_{CI}$ and $\mathcal{P}_{CD}$ for the three soft constraints, i.e., for Cluster 5 (heights A and D) and Cluster 6 (height G) in Fig 3e. The $\mathcal{P}_{WE}$ for the measurements (Cluster 1 to Cluster 4) subject to the scenarios of three soft constraints (heights A, D and G) are displayed in Fig 12. Fig 13 gives $\mathcal{P}_{WE}$ for three constraints (i.e., heights A, D and G). The $\mathcal{P}_{over+}$, $\mathcal{P}_{over-}$ and $\mathcal{P}_{ol}$ were also practically null for that case of three soft constraints. The sensitivity indicators (MDB and MIB) for each scenario of three soft constraints are displayed in Table 6.

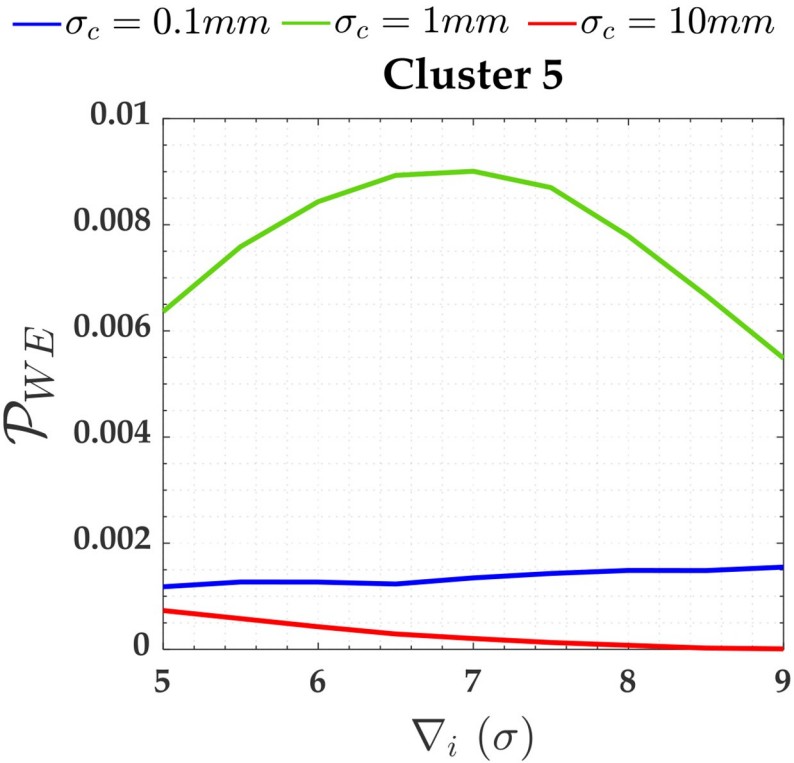

**Fig 9. The $\mathcal{P}_{WE}$ for the two soft constraints and for $\alpha' = 0.001$.** Cluster 5: heights A and D.

## Discussion

We started by analyzing the scenario of one hard constraint in Fig 3a. Table 1 shows that the maximum correlation between $w$-test statistics for the measurements constituting Cluster 1 is exactly equal to 1.00 (i.e., $max_{\rho_{w_i,w_j}} = 1.00$). This means that the measurements belonging to Cluster 1 are connected with unknown heights whose connections are limited to only two. Both unknown heights A and D are tied only to two measurements (i.e., $y_1$ and $y_6$ linked to A, and $y_3$ and $y_4$ linked to D); therefore, if an outlier occurred in one of these measurements, we would only be able to analyze the consistency between them, but we would not be able to distinguish which of them was contaminated by an outlier. This means that we would only be able to detect them, because the $w$-test statistics could be larger than a critical value $\hat{k}$; however,

**Table 4. MDB and MIB for the case of two soft constraints based on $\alpha' = 0.001$ and $\tilde{\mathcal{P}}_{CD} = \tilde{\mathcal{P}}_{CI} = 0.8$.**

| | $\sigma_c = 10mm$ | | $\sigma_c = 1mm$ | | $\sigma_c = 0.1mm$ | |
|---|---|---|---|---|---|---|
| Cluster | MDB ($\sigma$) | MIB ($\sigma$) | MDB ($\sigma$) | MIB ($\sigma$) | MDB ($\sigma$) | MIB ($\sigma$) |
| 1 | 7.5 | 25 | 7 | 7.1 | 6.3 | 6.3 |
| 2 | 6.8 | 6.8 | 6.6 | 6.6 | 6.3 | 6.3 |
| 3 | 6.4 | 6.4 | 6.4 | 6.4 | 6.3 | 6.3 |
| 4 | 6.3 | 6.3 | 6.3 | 6.3 | 6.3 | 6.3 |
| 5 | 6.8 | - | 8.8 | - | 57 | - |

**Table 5. Local redundancy ($r_i$), standard deviation of the LS-estimated outlier $\sigma_{\nabla_i}$ (mm) and the maximum absolute correlation ($max_{\rho_{w_i.w_j}}$) for each scenario of the three soft constraints.**

| Measurement | $\sigma_c = 0.1mm$ | | | $\sigma_c = 1mm$ | | | $\sigma_c = 10mm$ | | |
|---|---|---|---|---|---|---|---|---|---|
| | $r_i$ | $\sigma_{\nabla_i}$ | $max_{\rho_{w_i.w_j}}$ | $r_i$ | $\sigma_{\nabla_i}$ | $max_{\rho_{w_i.w_j}}$ | $r_i$ | $\sigma_{\nabla_i}$ | $max_{\rho_{w_i.w_j}}$ |
| $y_1$ | 0.702 | 1.194 | 0.660 | 0.502 | 1.411 | 0.577 | 0.398 | 1.586 | 0.992 |
| $y_2$ | 0.582 | 1.311 | 0.326 | 0.533 | 1.369 | 0.412 | 0.501 | 1.413 | 0.470 |
| $y_3$ | 0.702 | 1.194 | 0.660 | 0.502 | 1.411 | 0.577 | 0.398 | 1.586 | 0.992 |
| $y_4$ | 0.702 | 1.194 | 0.660 | 0.502 | 1.411 | 0.577 | 0.398 | 1.586 | 0.992 |
| $y_5$ | 0.582 | 1.311 | 0.326 | 0.533 | 1.369 | 0.412 | 0.501 | 1.413 | 0.470 |
| $y_6$ | 0.702 | 1.194 | 0.660 | 0.502 | 1.411 | 0.577 | 0.398 | 1.586 | 0.992 |
| $y_7$ | 0.704 | 1.192 | 0.415 | 0.602 | 1.289 | 0.412 | 0.563 | 1.333 | 0.470 |
| $y_8$ | 0.704 | 1.192 | 0.415 | 0.602 | 1.289 | 0.412 | 0.563 | 1.333 | 0.470 |
| $y_9$ | 0.704 | 1.192 | 0.415 | 0.602 | 1.289 | 0.412 | 0.563 | 1.333 | 0.470 |
| $y_{10}$ | 0.704 | 1.192 | 0.415 | 0.602 | 1.289 | 0.412 | 0.563 | 1.333 | 0.470 |
| $y_{11}$ | 0.583 | 1.309 | 0.326 | 0.583 | 1.309 | 0.385 | 0.583 | 1.309 | 0.433 |
| $y_{12}$ | 0.583 | 1.309 | 0.326 | 0.583 | 1.309 | 0.385 | 0.583 | 1.309 | 0.433 |
| $y_{13}$ | 0.012 | 0.904 | 0.660 | 0.425 | 1.534 | 0.542 | 0.663 | 12.283 | 0.501 |
| $y_{14}$ | 0.012 | 0.904 | 0.660 | 0.425 | 1.534 | 0.542 | 0.663 | 12.283 | 0.501 |
| $y_{15}$ | 0.019 | 0.718 | 0.63 | 0.5 | 1.414 | 0.542 | 0.665 | 12.268 | 0.501 |

in that case, the values of $w$-test statistics would be the same, and we would not have only one unique maximum $w$-test statistics, but would actually have four maximum $w$-test statistics. In other words, the equation systems associated with the measurements of Cluster 1 are linearly dependent [65]; therefore, there is no reliability in terms of outlier identification for Cluster 1, as can be seen in Fig 3a.

From Fig 3b, we note that there is reliability in terms of outlier detection for Cluster 1, and it is caused by overlapping $w$-test statistics. The probability of statistics overlap ($\mathcal{P}_{ol}$) for Cluster 1 in the scenario of a minimally constrained network is displayed in Fig 14.

The problem of not having more connections (i.e., more measurements) for the unknown heights A and D in the case of one hard constraint with G fixed is overcome when these heights (A and D) are taken as hard constraints in Fig 3b or when the heights A, D and G are hard constraints in Fig 3c. Fig 3a and 3b show that the measurements of Cluster 1 are able to identify an outlier when two hard constraints (A and D fixed) are in play. The case of three hard constraints (A, D and G fixes) in Fig 3e and 3f is also verified by our results i.e., there is reliability in terms of both outlier detection and identification for these measurements in those conditions.

From Table 2, we observe different behavior for the clusters as follows:

- Cluster 1: there was no MIB for the case of having only one single hard constraint, whereas there was $MDB = MIB$ for the other cases; however, both MDB and MIB decrease significantly with the increase in the number of hard constraints.

- Cluster 2: MDB was slightly smaller than MIB. Both MDB and MIB were practically the same for the case of having two or three hard constraints.

- Cluster 3: $MDB = MIB$ for all cases of hard constraints; however, both MDB and MIB decrease significantly with the increase in the number of hard constraints.

- Cluster 4: MDB and MIB were equal for all cases.

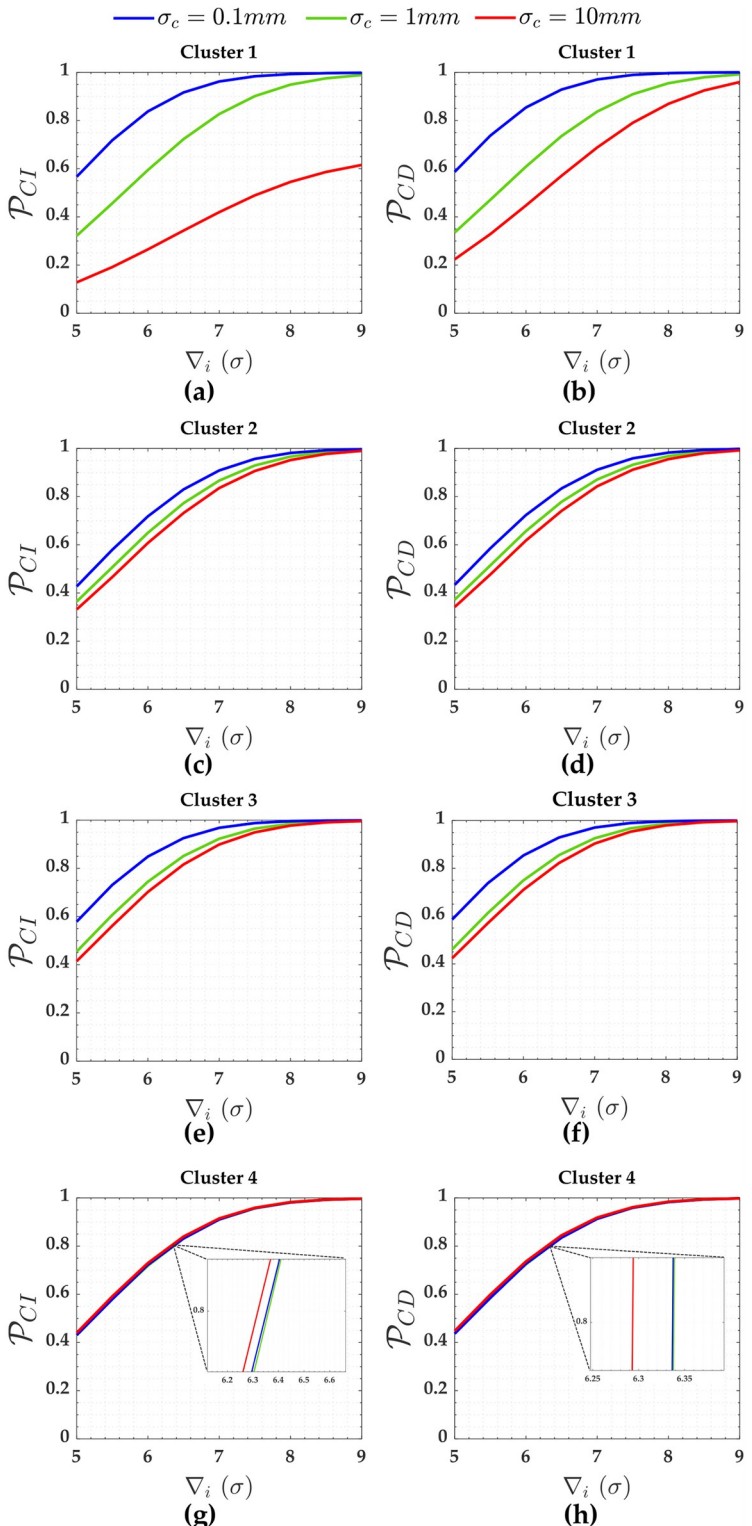

**Fig 10. The $\mathcal{P}_{CI}$ and $\mathcal{P}_{CD}$ for the measurements subject to the scenarios of three soft constraints for $\alpha' = 0.001$.** Cluster 1(**A**,**b**), Cluster 2(**c**,**d**), Cluster 3(**e**,**f**) and Cluster 4(**g**,**h**).

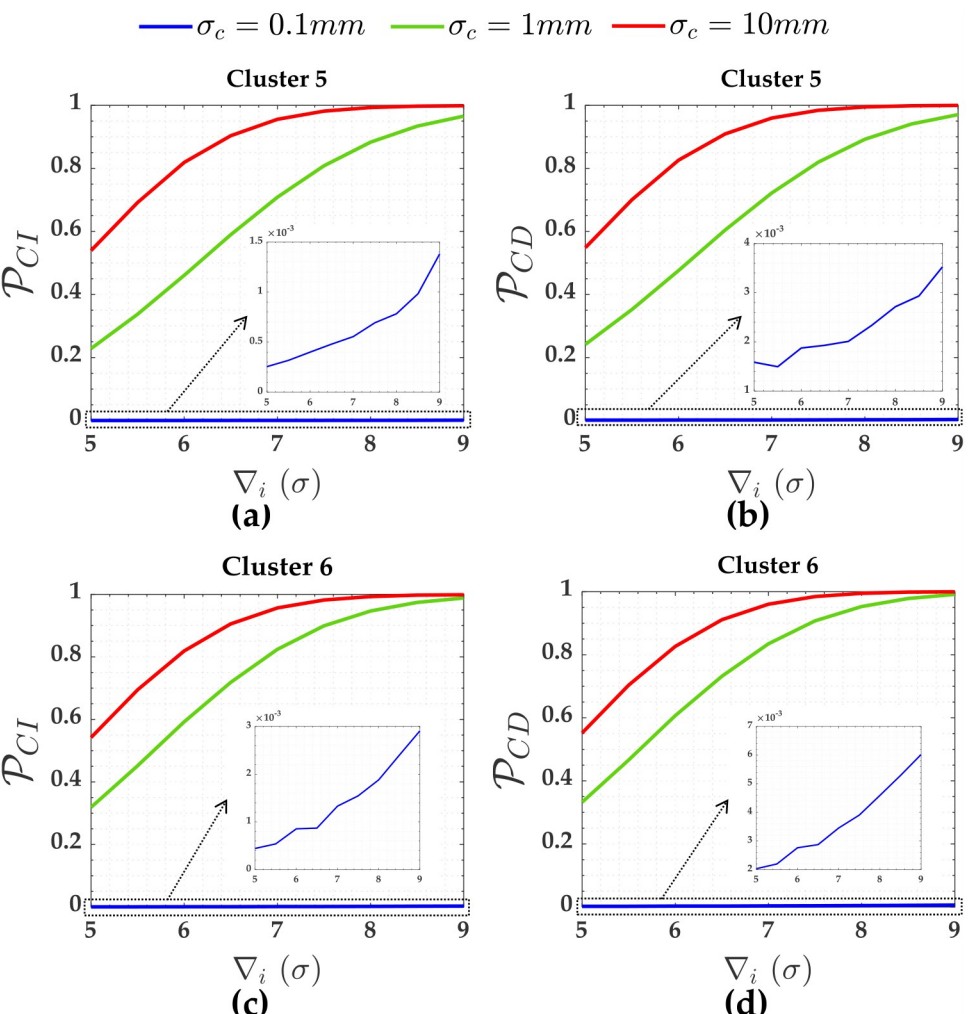

**Fig 11. The $\mathcal{P}_{CI}$ and $\mathcal{P}_{CD}$ for the three constraints and for $\alpha' = 0.001$.** Cluster 5(**a,b**) and Cluster 6(**c,d**).

In terms of outlier detection and identification: Cluster 1 was more sensitive to constraints; Cluster 3 was relatively sensitive to constraints; Cluster 4 was completely insensitive to constraints; Cluster 2 was relatively insensitive to constraints; see Fig 4. The reason for this is that the local redundancy ($r_i$) of Cluster 1 increased with the increase of the number of hard constraints, whereas Cluster 4 remained the same; see Table 2.

Leaving aside the cases of $\mathcal{P}_{ol}$, the network presents low least-squares residuals correlation ($\rho_{w_i,w_j} < 0.5$) and high local redundancy ($r_i > 0.5$). Because of this, $\mathcal{P}_{WE}$ were less than 1%, see Fig 5. The $\mathcal{P}_{over+}$ and $\mathcal{P}_{over-}$ were practically null. Consequently, $\mathcal{P}_{CI} \approx \mathcal{P}_{CD}$. Due of this fact, the family-wise error rate ($\alpha'$) should be increased in order to have more success rate in the outlier detection and identification [44].

From Fig 15, we observe that increasing the $\alpha'$ increases both the $\mathcal{P}_{CI}$ and $\mathcal{P}_{CD}$ for outlier magnitude from $5\sigma$ to $6\sigma$ in the case of three hard constraints and from $5\sigma$ to $6.8\sigma$ in the case of two hard constraints. Although the rates of $\mathcal{P}_{over+}$ and $\mathcal{P}_{WE}$ also increase, they are not

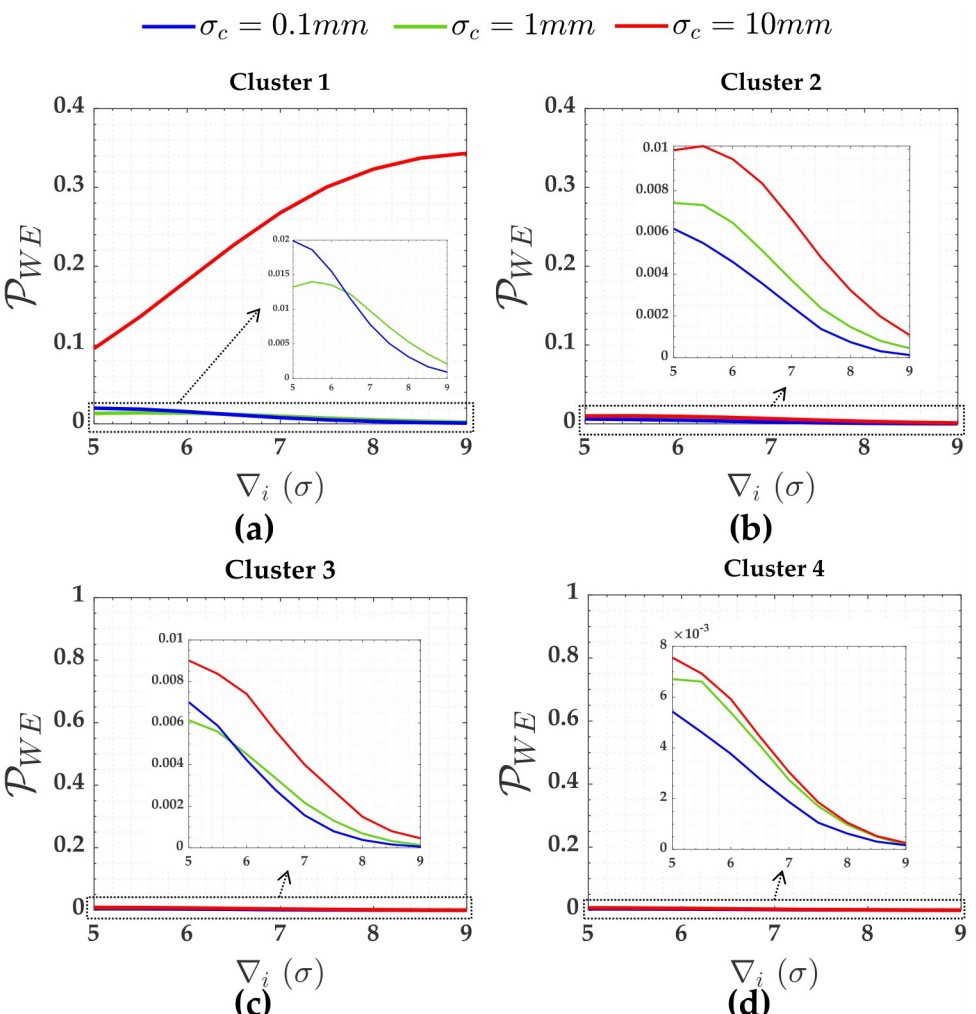

**Fig 12. The $\mathcal{P}_{WE}$ for the measurements subject to the scenarios of the three soft constraints and for $\alpha' = 0.001$.** Cluster 1(**a**), Cluster 2(**b**), Cluster 3(**c**) and Cluster 4(**d**).

significant when compared to the improvement of $\mathcal{P}_{CI}$ and detection ($\mathcal{P}_{CD}$). This same analysis can be done for the other clusters.

In terms of soft constraints for the cases of two constraints in Fig 3d, we observe from Table 3 that the larger the relaxation of the constraint (i.e., the larger the standard deviation of the constraint $\sigma_c$), the larger the residuals correlation ($\rho_{w_i,w_j}$) and the standard deviation of the outlier $\sigma_{\nabla_i}$, and the smaller the local redundancy ($r_i$). Consequently, $\mathcal{P}_{CI}$ and detection ($\mathcal{P}_{CD}$) get smaller and smaller with the relaxation of the constraints, whereas $\mathcal{P}_{WE}$ gets larger ($\mathcal{P}_{WE}$). This can be more clearly verified in Fig 6a, 6b and 8a for Cluster 1, whose measurements are connected with the constraints A and D (i.e., $y_{13}$ and $y_{14}$ in Table 3, respectively).

Note from Fig 8a that the $\mathcal{P}_{WE}$ increases as the magnitude of the outlier ($\nabla_i$) increases; however, this is only true up to a certain limit of outlier magnitude. The effect of residuals correlation $\rho_{w_i,w_j}$ on the rates of $\mathcal{P}_{WE}$ and $\mathcal{P}_{CI}$ tends to decrease with the increase in the magnitude of

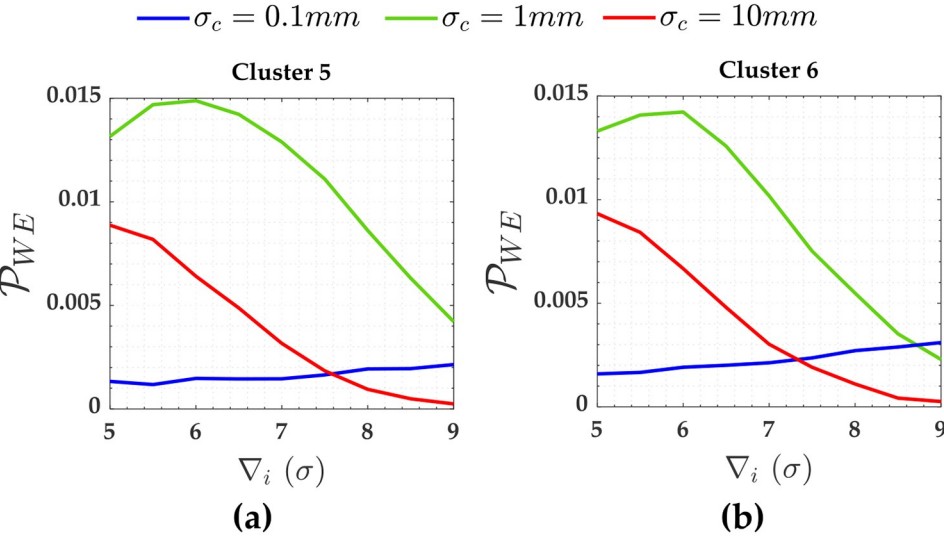

**Fig 13. The $\mathcal{P}_{WE}$ for the three constraints and for $\alpha' = 0.001$.** Cluster 6(**b**).

the outlier $\nabla_i$. This effect is more clearly verified for Cluster 1, in a case where the precision of the constraints are ten times worse than the measurements $\sigma_c = 10\sigma = 10mm$.

Note from Fig 6 that identifying an outlier in Cluster 1 (i.e., $y_1$, $y_3$, $y_4$ and $y_6$) when $\sigma_c = 10mm$ is more difficult than the other clusters. This is due to the fact that Cluster 1 has a higher residuals correlation $\rho_{w_i,w_j} = 0.994$ than other clusters. We observe that the larger the relaxation of the constraints, the larger the effect of the correlation $\rho_{w_i,w_j}$ on the success rate of outlier identification ($\mathcal{P}_{CI}$). Consequently, the higher the sensitivity indicator for outlier identification (MIB). Table 2 reveals that the ratio between MIB and MDB for Cluster 1 and for the scenario where the standard deviations of that two soft constraints are $\sigma_c = 10mm$ is MIB/MDB = 25/7.5 = 3.3. On the other hand, the relationship between MIB and MDB is practically one (i.e., MIB/MDB = 1.0) for the others scenarios.

If the *family-wise error rate* (FWE) rate ($\alpha'$) were increased for the case where the two soft constraints of $\sigma_c = 10mm$ are in play, we would not have great advantages for Cluster 1, due to its high residuals correlation ($\rho_{w_i,w_j} = 99.4\%$). From Fig 16, we can observe that the $\mathcal{P}_{CI}$ for outlier magnitudes from $5\sigma$ to $8\sigma$ is effectively larger for a user-defined $\alpha' = 0.1$ than $\alpha' = 0.001$; however, the success rate is still less than 80%, i.e., $\mathcal{P}_{CI} < 0.8$. Note, for example, the correct

**Table 6. MDB and MIB for the case of the three soft constraints based on $\alpha' = 0.001$ and $\tilde{\mathcal{P}}_{CD} = \tilde{\mathcal{P}}_{CI} = 0.8$.**

| Cluster | $\sigma_c = 10mm$ | | $\sigma_c = 1mm$ | | $\sigma_c = 0.1mm$ | |
|---|---|---|---|---|---|---|
| | MDB ($\sigma$) | MIB ($\sigma$) | MDB ($\sigma$) | MIB ($\sigma$) | MDB ($\sigma$) | MIB ($\sigma$) |
| 1 | 7.5 | 22 | 6.8 | 6.9 | 5.8 | 5.9 |
| 2 | 6.8 | 6.9 | 6.6 | 6.7 | 6.4 | 6.4 |
| 3 | 6.4 | 6.4 | 6.3 | 6.3 | 5.8 | 5.8 |
| 4 | 6.3 | 6.3 | 6.3 | 6.3 | 6.3 | 6.3 |
| 5 | 5.9 | 6.0 | 7.4 | 7.5 | 43.5 | 45 |
| 6 | 5.9 | 5.9 | 6.9 | 6.9 | 34.6 | 35.5 |

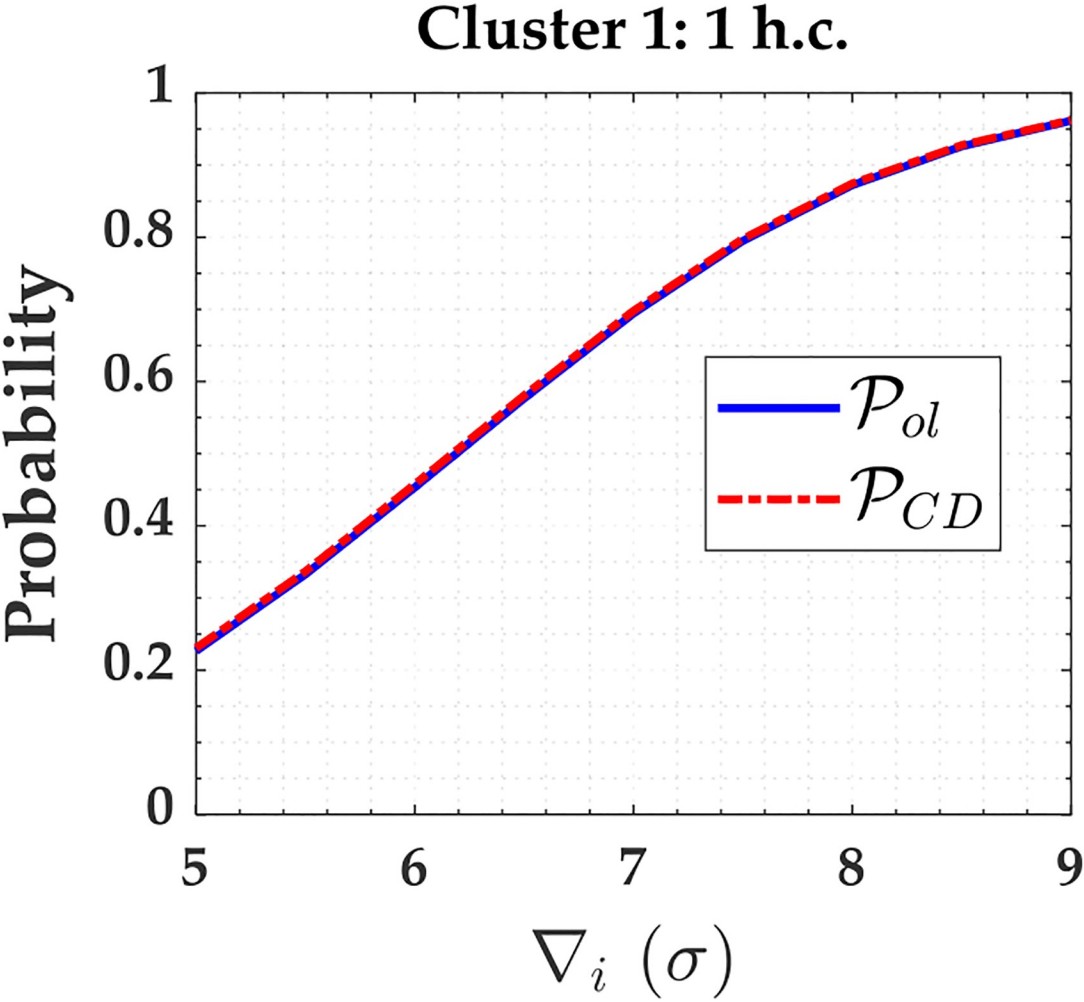

**Fig 14. $\mathcal{P}_{CD}$ and $\mathcal{P}_{ol}$ for Cluster 1 subject to one hard constraint and for $\alpha' = 0.001$.** The $\mathcal{P}_{CD}$ and $\mathcal{P}_{ol}$ for Cluster 1 subject to one hard constraint and for $\alpha' = 0.001$.

identification rate is $\mathcal{P}_{CI} = 56\%$ for an outlier magnitude of $\nabla_i = 8\sigma$ and $\alpha' = 0.1$. For $\alpha' = 0.1$ the $MIB = 33.5\sigma = 33.5mm$, whereas for $\alpha' = 0.001$ is $MIB = 25\sigma = 25mm$; therefore, in that case, the MIB for $\mathcal{P}_{CI} = 0.8(80\%)$ and $\alpha' = 0.1$ would be 34% larger than user-defined $\alpha' = 0.001$.

The soft constraints A and D were grouped in Cluster 5 (i.e., A and D were treated as pseudo-observations in the model). There is no reliability in terms of outlier identification for the constraints, because the residual correlation between them is $\rho_{w_i,w_j} = 100\%$, as can be seen in Table 3 for $y_{13}$ and $y_{14}$; however, these soft constraints are able to detect an outlier. In that case, the $\mathcal{P}_{CD}$ in Fig 7 is mainly caused by the $\mathcal{P}_{ol}$, as can be seen in $\sigma_c = 10mm$ in Fig 17. From Table 4, we observe that the larger the relaxation of the constraints, the larger the MDB. Note that the values of MDB are given in $\sigma$, and thus the MDB for $\sigma_c = 10mm$ is larger than $\sigma_c = 1mm$ and $\sigma_c = 0.1mm$, i.e., we had the following inequality: $MDB = 6.8\sigma_c = 6.8 \times 10mm = 68mm > MDB = 8.8\sigma_c = 8.8 \times 1mm = 8.8mm > MDB = 57\sigma_c = 57 \times 0.1mm = 5.7mm$. In that

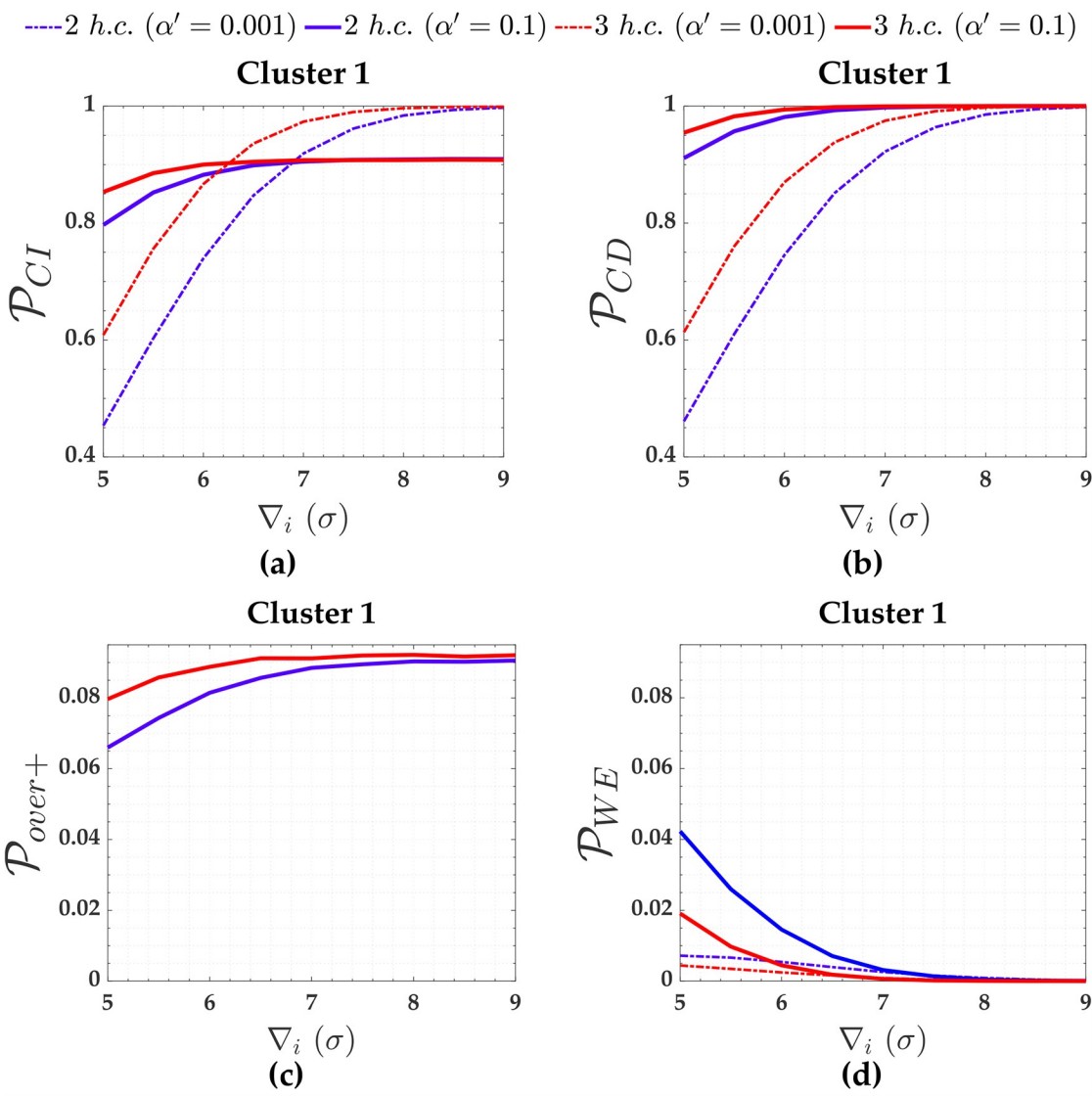

**Fig 15. The** $\mathcal{P}_{CI}$, $\mathcal{P}_{CD}$, $\mathcal{P}_{over+}$ **and** $\mathcal{P}_{WE}$ **for Cluster 1 subject to two and three hard constraints and for** $\alpha' = 0.001$ **and** $\alpha' = 0.1$. The $\mathcal{P}_{CI}$ (**A**), $\mathcal{P}_{CD}$ (**b**), $\mathcal{P}_{over+}$ (**c**) and $\mathcal{P}_{WE}$ for Cluster 1 subject to two and three hard constraints and for $\alpha' = 0.001$ and $\alpha' = 0.1$.

case, if the FWE ($\alpha'$) were increased, the rate of outlier detection by the Cluster 4 (i.e., by the soft constraints) would increase.

Similar effects of the relaxation of the constraints on the performance of the *IDS* in case of two soft constraints are verified in case of three soft constraints, as can be seen in Figs 10, 11, 12 and 13.

In case of having three soft constraints in Fig 3e, there is reliability in terms of outlier identification for the three pseudo-observations $y_{13}$, $y_{14}$ and $y_{15}$ (i.e., for A, D and G), seen in Fig 10 and Table 6. In that case, we also observe that $\mathcal{P}_{CD}$ of the soft constraints A and D (i.e., Cluster 5) were approximately 13% for $\sigma_c = 10mm$, 16% for $\sigma_c = 1mm$ and 24% for $\sigma_c = 0.1mm$ larger than the scenario of the network subject to two soft constraints. Table 6 reveals that the advantage of having three soft constraints instead of two constraints is that the constraints become identifiable in the presence of an outlier. The behavior of the $\mathcal{P}_{CD}$, $\mathcal{P}_{CI}$ and $\mathcal{P}_{WE}$ was similar to

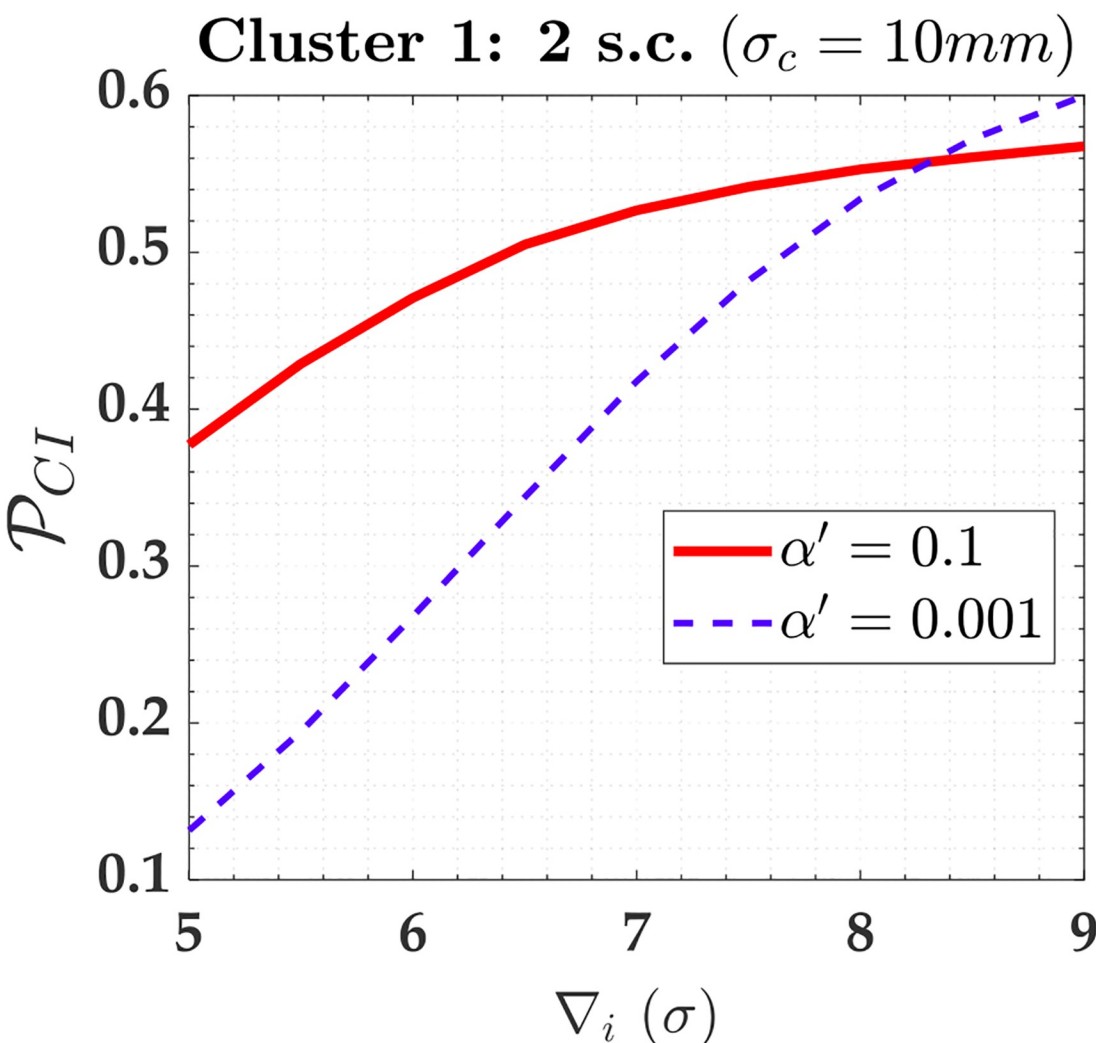

**Fig 16. The** $\mathcal{P}_{CI}$ **for Cluster 1 subject to two soft constraints (2 s.c.) A and D for** $\alpha' = 0.001$ **and** $\alpha' = 0.1$. The $\mathcal{P}_{CI}$ for Cluster 1 subject to two soft constraints (2 s.c.) A and D for $\alpha' = 0.001$ and $\alpha' = 0.1$.

the case of the two soft constraints. Furthermore, the larger the relaxation of the constraints, the smaller the residuals correlation between the measurements and the soft constraints and the larger the residuals correlation among the measurements.

We also observe that the case of two soft constraints for $\sigma_c = 0.1mm$ was comparable with two hard constraints (see e.g., Tables 2 and 6) in terms of the probability levels associated with *IDS* for the measurements (i.e., clusters 1, 2, 3 and 4). In the same way for the case of two soft constraints with $\sigma_c = 1mm$ or $\sigma_c = 10mm$, the probabilities levels were similar to the one hard constraint for that measurements, with the benefit of two soft constraints having reliability in terms of outlier identification for the Cluster 1. Finally, the three soft constraints with $\sigma_c = 1mm$ and $\sigma_c = 10mm$ were comparable to the two soft constraints for that scenario of constraints relaxation, wheres the three soft constraints for $\sigma_c = 0.1mm$ showed similar outcomes with three hard constraints for the measurements (see e.g., Tables 2 and 6). In that case, however, an advantage of the three soft constraints on the three hard constraints is the possibility of analyzing the sensitivity of the constraints. We emphasize that the stochastic models of the

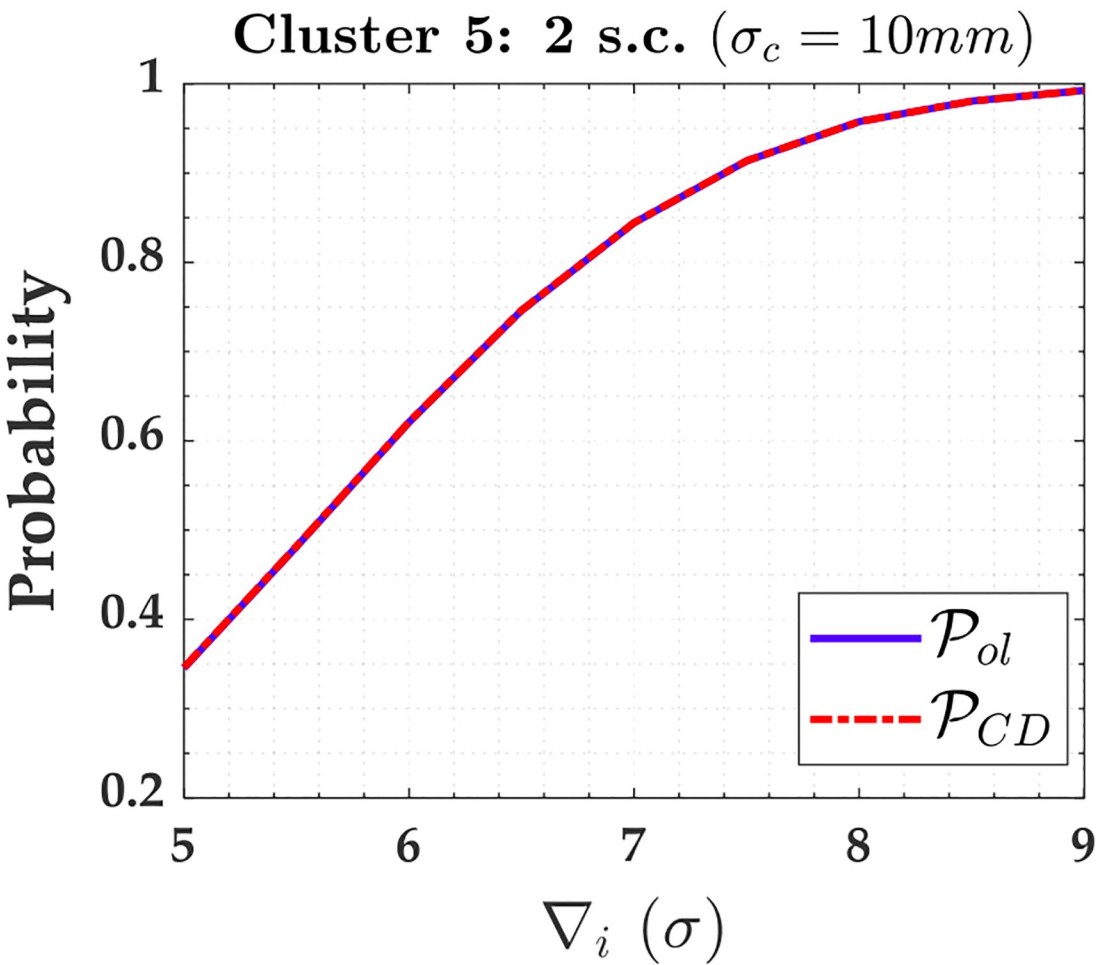

**Fig 17. The $\mathcal{P}_{CD}$ and $\mathcal{P}_{ol}$ for the two soft constraints A and D (Cluster 5) with $\sigma_c = 10mm$ and for $\alpha' = 0.001$.** The $\mathcal{P}_{CD}$ and $\mathcal{P}_{ol}$ for the two soft constraints A and D (Cluster 5) with $\sigma_c = 10mm$ and for $\alpha' = 0.001$.

measurements and constraints were assumed to be well-known and defined for the analyses performed here.

## Conclusion

We highlight the main findings of this research as follows:

- Under a system of a high local redundancy $r_i > 0.5$ and low residuals correlation ($\rho_{w_i,w_j} < 0.5$), if one increases the *family-wise error rate* (FWE) of the test statistic, the performance of the procedure will be improved for both scenarios of hard constraints and soft constraints.

- $\mathcal{P}_{CI}$ of the observations is larger for the case of hard constraints than soft constraints.

- The larger the relaxation of the constraints, the larger the effect of the residuals correlation ($\rho_{w_i,w_j}$) on the success rate of outlier identification ($\mathcal{P}_{CI}$) of the observations. Consequently, the higher the sensitivity indicator for outlier identification (MIB), the more difficult it becomes to identify an outlier.

- Under a scenario of soft constraints, one should set out at least three soft constraints in order to identify an outlier in the constraints.

- Hard constraints should be used in the stage of pre-processing data for the purpose of identifying and removing possible outlying measurements. In that process, one should opt to set out the redundant hard constraints at points in the network where the smallest connections exist. After identifying and removing possible outliers, the soft constraints should be employed to propagate the uncertainties of the constraints (pseudo-observations) to the model parameters during the process of least-squares estimation.

## Supporting information

**S1 Appendix. Description of the method.** Provides a broad theoretical framework and detailed description of the method used to estimate the Iterative Data-Snooping probability levels.
(PDF)

## Acknowledgments

The authors would like to thank the two anonymous reviewers who contributed to the improvement of the manuscript.

## Author Contributions

**Conceptualization:** Vinicius Francisco Rofatto, Marcelo Tomio Matsuoka, Ivandro Klein.

**Data curation:** Vinicius Francisco Rofatto, Ivandro Klein.

**Formal analysis:** Vinicius Francisco Rofatto.

**Funding acquisition:** Marcelo Tomio Matsuoka, Maurício Roberto Veronez, Luiz Gonzaga da Silveira, Junior.

**Investigation:** Vinicius Francisco Rofatto.

**Methodology:** Vinicius Francisco Rofatto, Marcelo Tomio Matsuoka, Ivandro Klein.

**Project administration:** Marcelo Tomio Matsuoka, Maurício Roberto Veronez.

**Software:** Vinicius Francisco Rofatto, Luiz Gonzaga da Silveira, Junior.

**Supervision:** Marcelo Tomio Matsuoka, Ivandro Klein, Maurício Roberto Veronez, Luiz Gonzaga da Silveira, Junior.

**Validation:** Vinicius Francisco Rofatto, Ivandro Klein, Maurício Roberto Veronez, Luiz Gonzaga da Silveira, Junior.

**Writing – original draft:** Vinicius Francisco Rofatto.

**Writing – review & editing:** Vinicius Francisco Rofatto, Marcelo Tomio Matsuoka, Ivandro Klein, Maurício Roberto Veronez, Luiz Gonzaga da Silveira, Junior.

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
