## [Decision Letter · Decision Letter 0]

29 Jun 2020

PONE-D-20-10931

On the Effects of Hard and Soft Equality Constraints in the Iterative Outlier Elimination Procedure

PLOS ONE

Dear Dr. ROFATTO,

Thank you for submitting your manuscript to PLOS ONE. After careful consideration, we feel that it has merit but does not fully meet PLOS ONE’s publication criteria as it currently stands. Therefore, we invite you to submit a revised version of the manuscript that addresses the points raised during the review process.

 Based on the comments from the reviewers, a minor revision is needed for improving the quality of the manuscript. Some details should be highlighted in the revised version. Meanwhile, a proof reading is recommenced before submitting the revised version. Please consider and response all the comments of the reviewer, a separate response letter is essential.

We look forward to receiving your revised manuscript.

Kind regards,

Qichun 'Kit' Zhang, PhD, FHEA, CEng, MIET, SMIEEE

Academic Editor

PLOS ONE

University of Bradford

Journal Requirements:

'The authors would like to thank CNPq|Conselho Nacional de Desenvolvimento Cientfico e Tecnologico|Brasil (proc. nº 103587/2019-5) and PETROBRAS (Grant Number 2018/00545-0) for funding the research.'

'The funders had no role in study design, data collection and analysis, decision to publish, or preparation of the manuscript.'

Reviewers' comments:

Reviewer's Responses to Questions

**Comments to the Author**

1. Is the manuscript technically sound, and do the data support the conclusions?

Reviewer #1: Yes

Reviewer #2: Yes

2. Has the statistical analysis been performed appropriately and rigorously? 

Reviewer #1: Yes

Reviewer #2: Yes

3. Have the authors made all data underlying the findings in their manuscript fully available?

Reviewer #1: Yes

Reviewer #2: Yes

4. Is the manuscript presented in an intelligible fashion and written in standard English?

Reviewer #1: Yes

Reviewer #2: Yes

5. Review Comments to the Author

Reviewer #1: The study “On the Effects of Hard and Soft Equality Constraints in the Iterative Outlier Elimination Procedure” is interesting. The paper is well set. The data and methodology parts are well described. However, I would recommend the authors add some more explanations and also describe the innovation of this paper in the introduction section. Moreover, attention should be given to the following highlighted points before resubmitting.

1. Improve the quality of Figure 1. Flowchart of the algorithm because the text inside the boxes is hard to read.

2. Page 3 / 20, Line 85, the words “ the minimal biases, MDB (Minimal Detectable Bias) and MIB (Minimal Identifiable Bias) “, the same words defined again just with a different style as Page 4 / 20 line 108 “ the minimal biases - Minimal Detectable Bias (MDB) and Minimal Identifiable Bias (MIB) ”, Please use the same style throughout the paper and secondly once an abbreviation is defined just use the same if required. The same repeated on page 7 / 20 lines 227 MIB (Minimal Identifiable Bias).

3. Page 6 / 20 line 168. The expression 1/0=∞ is not true. 1/0 is said to be undefined because the division is defined in terms of multiplication. a/b = x is defined to mean that b*x = a. There is no x such that 0*x = 1, since 0*x = 0 for all x. Thus 1/0 does not exist, or is not defined, or is undefined.

4. In Table 1 with two hard constraints, why the local redundancy, the standard deviation least square estimated outlier, and the maximum absolute correlation all are equal for twelve measurements. While for 1 hard constraint and 3 hard constraints there is some variation present in twelve measurements.

5. Page 9 / 20. The abbreviations used are very much common please stop this work from line 254 to 259, probabilities of correct identification (PCI ) and correct detection (PCD) used and defined 3 and 4 times, respectively. The other abbreviations also repeated quite often i.e. over-identification cases (P over+ and P over-) and the statistical overlap (Pol).

6. Check the value of MIB in Table 4, column 3 is it the value equal to 25?

7. Check the value of MIB in Table 6, column 3 is it the value equal to 22?

8. Page 12 / 20 line 324, Cluster 4: MDB e MIB were equal for all cases. Here the lower case e stands for.

9. In last some special attention must be given to the language as well. There are some sentences which completely wrong.

Reviewer #2: The authors investigated the effect of soft and hard constraints in the iterative outlier elimination procedure.

The paper is well written and addressing an important issue in the field of Statistical Process Control. The paper can be acceptable for publication after careful handling of the following points.

i) Reduce the length of Conclusion section. Only include important findings.

ii) Include high quality figures.

iii) There are some missing lines or extra legend items in Figures 4(c) - 4(h).

iv) Figures should be of same size.

6. PLOS authors have the option to publish the peer review history of their article (what does this mean?). If published, this will include your full peer review and any attached files.

Reviewer #1: No

Reviewer #2: No

---

## [Author Response · Author response to Decision Letter 0]

17 Jul 2020

Editor: We have carefully reviewed the comments and thoroughly revised the manuscript accordingly. We would like to update our Funding Statement, as follows:

 - The CNPq - Conselho Nacional de Desenvolvimento Científico e Tecnológico - Brasil had the role of providing the study grant for the second author (proc. nº103587/2019-5); and

- The PETROBRAS (Grant Number 2018/00545-0) had the role of paying both the publication fee and the professional language editing service. 

We have provided the following files as requested:

English language editing certificate by MDPI.

A copy of our manuscript showing our changes by highlighting them.

A clean copy of the edited manuscript.

Reviewer 1: We have incorporated all of your suggestions into our revision. They were very helpful. Thank you. 

Reviewer 2: We have incorporated all of your suggestions into our revision. They were very helpful. Thank you for your help.

---

## [Decision Letter · Decision Letter 1]

11 Aug 2020

On the Effects of Hard and Soft Equality Constraints in the Iterative Outlier Elimination Procedure

PONE-D-20-10931R1

Dear Dr. ROFATTO,

We’re pleased to inform you that your manuscript has been judged scientifically suitable for publication and will be formally accepted for publication once it meets all outstanding technical requirements.

Kind regards,

Qichun Kit Zhang, PhD

Academic Editor

PLOS ONE

Additional Editor Comments:

Both reviewers satisfied the current version with revisions. The concerns have been addressed well and the quality of the manuscript has been improved. The paper is acceptable and ready to publish.

Reviewers' comments:

Reviewer's Responses to Questions

**Comments to the Author**

1. If the authors have adequately addressed your comments raised in a previous round of review and you feel that this manuscript is now acceptable for publication, you may indicate that here to bypass the “Comments to the Author” section, enter your conflict of interest statement in the “Confidential to Editor” section, and submit your "Accept" recommendation.

Reviewer #1: All comments have been addressed

Reviewer #2: All comments have been addressed

2. Is the manuscript technically sound, and do the data support the conclusions?

Reviewer #1: Yes

Reviewer #2: Yes

3. Has the statistical analysis been performed appropriately and rigorously? 

Reviewer #1: Yes

Reviewer #2: Yes

4. Have the authors made all data underlying the findings in their manuscript fully available?

Reviewer #1: Yes

Reviewer #2: Yes

5. Is the manuscript presented in an intelligible fashion and written in standard English?

Reviewer #1: Yes

Reviewer #2: Yes

6. Review Comments to the Author

Reviewer #1: As my suggestions form the previous review round have been met, I suggest acceptance of the paper in the current form.

Reviewer #2: All comments have been addressed in the revised submission.The manuscript is acceptable in the current format.

7. PLOS authors have the option to publish the peer review history of their article (what does this mean?). If published, this will include your full peer review and any attached files.

Reviewer #1: No

Reviewer #2: **Yes: **Saddam Akber Abbasi

---

## [Editor Report · Acceptance letter]

14 Aug 2020

PONE-D-20-10931R1 

On the Effects of Hard and Soft Equality Constraints in the Iterative Outlier Elimination Procedure 

Dear Dr. Rofatto:

I'm pleased to inform you that your manuscript has been deemed suitable for publication in PLOS ONE. Congratulations! Your manuscript is now with our production department. 

Kind regards, 

on behalf of

Dr. Qichun Zhang 

Academic Editor

PLOS ONE